# ERASEDIFF: ERASING DATA INFLUENCE IN DIFFUSION MODELS

## ABSTRACT

In response to data protection regulations and the "right to be forgotten", in this work, we introduce an unlearning algorithm for diffusion models. Our algorithm equips a diffusion model with a mechanism to mitigate the concerns related to data memorization. To achieve this, we formulate the unlearning problem as a bi-level optimization problem, wherein the outer objective is to preserve the utility of the diffusion model on the remaining data. The inner objective aims to scrub the information associated with forgetting data by deviating the learnable generative process from the ground-truth denoising procedure. To solve the resulting bi-level problem, we adopt a first-order method, having superior practical performance while being vigilant about the diffusion process and solving a bi-level problem therein. Empirically, we demonstrate that our algorithm can preserve the model utility, effectiveness, and efficiency while removing across two widely-used diffusion models and in both conditional and unconditional image generation scenarios. In our experiments, we demonstrate the unlearning of classes, attributes, and even a race from face and object datasets such as UTKFace, CelebA, CelebA-HQ, and CIFAR10. The source code of our algorithm is available at `https://github.com/AnonymousUser-hello/DiffusionUnlearning`.

## 1 INTRODUCTION

Diffusion Models (Ho et al., 2020; Song et al., 2020; Rombach et al., 2022) are now the method of choice in deep generative models, owing to their high-quality output, stability, and ease of training procedure. This has facilitated their successful integration into commercial applications such as *midjourney*[1]. Unfortunately, the ease of use associated with diffusion models brings forth significant privacy risks. Studies have shown that these models can memorize and regenerate individual images from their training datasets (Somepalli et al., 2023a;b; Carlini et al., 2023). Beyond privacy, diffusion models are susceptible to misuse, capable of generating inappropriate digital content (Rando et al., 2022; Salman et al., 2023; Schramowski et al., 2023). They are also vulnerable to poison attacks (Chen et al., 2023b), allowing the generation of target images with specific triggers. These factors collectively pose substantial security threats. Moreover, the ability of diffusion models to emulate distinct artistic styles (Shan et al., 2023; Gandikota et al., 2023a) raises questions about data ownership and compliance with intellectual property and copyright laws.

In this context, individuals whose images are used for training might request the removal of their private data. In particular, data protection regulations like the European Union General Data Protection Regulation (GDPR) (Voigt & Von dem Bussche, 2017) and the California Consumer Privacy Act (CCPA) (Goldman, 2020) grant users the *right to be forgotten*, obligating companies to expunge data pertaining to a user upon receiving a request for deletion. These legal provisions grant data owners the right to remove their data from trained models and eliminate its influence on said models (Bourtoule et al., 2021; Guo et al., 2020; Golatkar et al., 2020; Mehta et al., 2022; Sekhari et al., 2021; Ye et al., 2022; Tarun et al., 2023b;a; Chen et al., 2023a).

A straightforward solution for unlearning is to retrain the model from scratch after excluding the data that needs to be forgotten. However, the removal of pertinent data followed by retraining diffusion models from scratch demands substantial resources and is often deemed impractical. Existing

---

[1]https://docs.midjourney.com/

research on efficient unlearning have primarily focused on classification problems Romero et al. (2007); Karasuyama & Takeuchi (2010); Cao & Yang (2015); Ginart et al. (2019); Bourtoule et al. (2021); Wu et al. (2020); Guo et al. (2020); Golatkar et al. (2020); Mehta et al. (2022); Sekhari et al. (2021); Chen et al. (2023a), and cannot be directly applied to diffusion models. Consequently, there is an urgent need for the development of methods capable of scrubbing data from diffusion models without necessitating complete retraining.

In this work, we propose a method to scrub the data information from the diffusion models without requiring training the whole system from scratch. Specifically, the proposed method *EraseDiff* formulates diffusion unlearning as a bi-level optimization problem, where the outer objective is to finetune the models with the remaining data for preserving the model utility and the inner objective aims to erase the influence of the forgetting data on the models by deviating the learnable reverse process from the ground-truth denoising procedure. Then, a first-order solution is adopted to solve the resulting problem. We benchmark *EraseDiff* on various scenarios, encompassing unlearning of classes on CIFAR10 (Krizhevsky et al., 2009) and races on UTKFace (Zhang et al., 2017) with conditional diffusion models, attributes on CelebA (Liu et al., 2015) as well as CelebA-HQ (Lee et al., 2020) with unconditional diffusion models. The results demonstrate that *EraseDiff* surpasses the baseline methods for diffusion unlearning across a range of evaluation metrics.

## 2 Background

In the following section, we outline the components of the models we evaluate, including Denoising Diffusion Probabilistic Models (DDPM) (Ho et al., 2020), denoising diffusion implicit models (DDIM) (Song et al., 2020), and classifier-free guidance diffusion models (Ho & Salimans, 2022). Throughout the paper, we denote scalars, and vectors/matrices by lowercase and bold symbols, respectively (e.g., $a$, $\boldsymbol{a}$, and $\boldsymbol{A}$).

**DDPM.** (1) Diffusion: DDPM gradually diffuses the data distribution $\mathbb{R}^d \ni \mathbf{x}_0 \sim q(\mathbf{x})$ into the standard Gaussian distribution $\mathbb{R}^d \ni \boldsymbol{\epsilon} \sim \mathcal{N}(\mathbf{0}, \mathbf{I}_d)$ with $T$ time steps, i.e., $q(\mathbf{x}_t|\mathbf{x}_{t-1}) = \mathcal{N}(\mathbf{x}_t; \sqrt{\alpha_t}\mathbf{x}_{t-1}, (1-\alpha_t)\mathbf{I}_d)$, where $\alpha_t = 1 - \beta_t$ and $\{\beta_t\}_{t=1}^T$ are the pre-defined variance schedule. Then we can express $\mathbf{x}_t$ as $\mathbf{x}_t = \sqrt{\bar{\alpha}_t}\mathbf{x}_0 + \sqrt{1-\bar{\alpha}_t}\boldsymbol{\epsilon}$, where $\bar{\alpha}_t = \prod_{i=1}^t \alpha_i$. (2) Training: A model with parameters $\boldsymbol{\theta} \in \mathbb{R}^n$, i.e., $\epsilon_{\boldsymbol{\theta}}(\cdot)$ is applied to learn the reverse process $p_{\boldsymbol{\theta}}(\mathbf{x}_{t-1}|\mathbf{x}_t) \approx q(\mathbf{x}_{t-1}|\mathbf{x}_t)$. Given $\mathbf{x}_0 \sim q(\mathbf{x})$ and time step $t \in [1, T]$, the simplified training objective is to minimize the distance between $\boldsymbol{\epsilon}$ and the predicted $\boldsymbol{\epsilon}_t$ given $\mathbf{x}_0$ at time $t$, i.e., $\|\boldsymbol{\epsilon} - \epsilon_{\boldsymbol{\theta}}(\mathbf{x}_t, t)\|$. (3) Sampling: after training the model, we could obtain the learnable backward distribution $p_{\boldsymbol{\theta}^*}(\mathbf{x}_{t-1}|\mathbf{x}_t) = \mathcal{N}(\mathbf{x}_{t-1}; \boldsymbol{\mu}_{\boldsymbol{\theta}^*}(\mathbf{x}_t, t), \boldsymbol{\Sigma}_{\boldsymbol{\theta}^*}(\mathbf{x}_t, t))$, where $\boldsymbol{\mu}_{\boldsymbol{\theta}^*}(\mathbf{x}_t, t) = \frac{\sqrt{\bar{\alpha}_{t-1}}\beta_t}{1-\bar{\alpha}_t}\mathbf{x}_0 + \frac{\sqrt{\alpha_t}(1-\bar{\alpha}_{t-1})}{1-\bar{\alpha}_t}\mathbf{x}_t$ and $\boldsymbol{\Sigma}_{\boldsymbol{\theta}^*}(\mathbf{x}_t, t) = \frac{(1-\bar{\alpha}_{t-1})\beta_t}{1-\bar{\alpha}_t}$. Then, given $\mathbf{x}_T \sim \mathcal{N}(\mathbf{0}, \mathbf{I}_d)$, $\mathbf{x}_0$ could be obtained via sampling from $p_{\boldsymbol{\theta}^*}(\mathbf{x}_{t-1}|\mathbf{x}_t)$ from $t = T$ to $t = 1$ step by step.

**DDIM.** DDIM could be viewed as using a different reverse process, i.e., $p_{\boldsymbol{\theta}^*}(\mathbf{x}_{t-1}|\mathbf{x}_t) = \mathcal{N}(\mathbf{x}_{t-1}; \boldsymbol{\mu}_{\boldsymbol{\theta}^*}(\mathbf{x}_t, t), \sigma_t^2\mathbf{I}_d)$, where $\boldsymbol{\mu}_{\boldsymbol{\theta}^*}(\mathbf{x}_t, t) = \sqrt{\bar{\alpha}_{t-1}}\mathbf{x}_0 + \sqrt{1-\bar{\alpha}_{t-1}-\sigma_t^2}\frac{\mathbf{x}_t - \sqrt{\bar{\alpha}_t}}{\sqrt{1-\bar{\alpha}_t}}$ and $\sigma_t^2 = \eta\frac{(1-\bar{\alpha}_{t-1})\beta_t}{1-\bar{\alpha}_t}$, $\eta \in [0, 1]$. A stride sampling schedule is adopted to accelerate the sampling process.

**Classifier-free guidance.** Classifier-free guidance is a conditioning method to guide diffusion-based generative models without an external pre-trained classifier. Model prediction would be $\epsilon_{\boldsymbol{\theta}}(\mathbf{x}_t, t, c)$, where $c$ is the input's corresponding label. The unconditional and conditional models are jointly trained by randomly setting $c$ to the unconditional class $\emptyset$ identifier with the probability $p_{uncond}$. Then, the sampling procedure would use the linear combination of the conditional and unconditional score estimates as $\boldsymbol{\epsilon}_t = (1 + w) \cdot \epsilon_{\boldsymbol{\theta}}(\mathbf{x}_t, t, c) - w \cdot \epsilon_{\boldsymbol{\theta}}(\mathbf{x}_t, t)$, and $w$ is the guidance scale that controls the strength of the classifier guidance.

## 3 Diffusion Unlearning

Let $\mathcal{D} = \{\mathbf{x}_i, c_i\}_i^N$ be a dataset of images $\mathbf{x}_i$ associated with label $c_i$ representing the class. $\mathcal{C} = \{1, \cdots, C\}$ denotes the label space where $C$ is the total number of classes and $c_i \in [1, C]$. We

split the training dataset $\mathcal{D}$ into the forgetting data $\mathcal{D}_f \subset \mathcal{D}$ and its complement, the remaining data $\mathcal{D}_r := \mathcal{D}_f^{\complement}$. The forgetting data has label space $\mathcal{C}_f \subset \mathcal{C}$, and the remaining label space is denoted as $\mathcal{C}_r := \mathcal{C}_f^{\complement}$.

### 3.1 TRAINING OBJECTIVE

Our goal is to scrub the information about the forgetting data $\mathcal{D}_f$ carried by the diffusion models while maintaining the model utility over the remaining data $\mathcal{D}_r$. To achieve this, we adopt different training objectives for $\mathcal{D}_r$ and $\mathcal{D}_f$ as follows.

**Remaining data $\mathcal{D}_r$.** For the remaining data $\mathcal{D}_r$, we finetune the diffusion models with the original objective by minimizing the variational bound on negative log-likelihood:

$$\min_{\boldsymbol{\theta}} \mathbb{E}_{\mathbf{x}_0 \sim \mathcal{D}_r}[-\log p_{\boldsymbol{\theta}}(\mathbf{x}_0)] \propto \min_{\boldsymbol{\theta}} \mathbb{E}_{\mathbf{x}_0 \sim \mathcal{D}_r} \left[ \sum_{t=2}^{T} \underbrace{\mathrm{KL}(q(\mathbf{x}_{t-1}|\mathbf{x}_t,\mathbf{x}_0)\|p_{\boldsymbol{\theta}}(\mathbf{x}_{t-1}|\mathbf{x}_t))}_{\mathcal{L}_{t-1}(\boldsymbol{\theta},\mathcal{D}_r)} \right], \quad (1)$$

where $\mathbf{x}_t = \sqrt{\bar{\alpha}_t}\mathbf{x}_0 + \sqrt{1-\bar{\alpha}_t}\boldsymbol{\epsilon}$ with $\boldsymbol{\epsilon} \in \mathcal{N}(\mathbf{0}, \mathbf{I}_d)$. Given a large $T$ and if $\alpha_T$ is sufficiently close to 0, $x_T$ would converge to a standard Gaussian distribution, so $p_\theta(\mathbf{x}_T) := \mathcal{N}(\mathbf{0}, \mathbf{I}_d)$. Eq. (1) aims to minimize the KL divergence between the ground-truth backward distribution $q(\mathbf{x}_{t-1}|\mathbf{x}_t, \mathbf{x}_0) = \mathcal{N}(\mathbf{x}_{t-1}; \tilde{\boldsymbol{\mu}}_t, \tilde{\beta}_t^2 \mathbf{I}_d)$ and the learnable backward distribution $p_{\boldsymbol{\theta}}(\mathbf{x}_{t-1}|\mathbf{x}_t)$. With $p_{\boldsymbol{\theta}}(\mathbf{x}_{t-1}|\mathbf{x}_t) = \mathcal{N}(\mathbf{x}_{t-1}; \boldsymbol{\mu}_{\boldsymbol{\theta}}(\mathbf{x}_t, t), \sigma_t^2 \mathbf{I}_d)$, then we define

$$\mathcal{F}(\boldsymbol{\theta}) := \mathcal{L}_{t-1}(\boldsymbol{\theta}, \mathcal{D}_r) = \mathbb{E}_{\mathbf{x}_0 \sim \mathcal{D}_r, \boldsymbol{\epsilon} \sim \mathcal{N}(\mathbf{0}, \mathbf{I}_d)} \left[ a \cdot \left\| \boldsymbol{\epsilon} - \boldsymbol{\epsilon}_{\boldsymbol{\theta}}(\sqrt{\bar{\alpha}_t}\mathbf{x}_0 + \sqrt{1-\bar{\alpha}_t}\boldsymbol{\epsilon}, t) \right\|^2 \right], \quad (2)$$

where the coefficient $a = \frac{\beta_t^2}{2\sigma_t^2 \alpha_t (1-\bar{\alpha}_t)}$ for DDPM and $a = \frac{(\sqrt{\alpha_t(1-\bar{\alpha}_{t-1}-\sigma_t^2)}-\sqrt{1-\bar{\alpha}_t})^2}{2\sigma_t^2 \alpha_t}$ for DDIM. Eq. (2) constraints the model $\boldsymbol{\epsilon}_{\boldsymbol{\theta}}$ to predict $\boldsymbol{\epsilon}$ from $\mathbf{x}_t$, with the goal of aligning the learnable backward distribution $p_{\boldsymbol{\theta}}(\mathbf{x}_{t-1}|\mathbf{x}_t)$ closely with the ground-truth backward distribution $q(\mathbf{x}_{t-1}|\mathbf{x}_t, \mathbf{x}_0)$.

**Forgetting data $\mathcal{D}_f$.** For the forgetting data $\mathcal{D}_f$, we update the approximator $\boldsymbol{\epsilon}_{\hat{\boldsymbol{\theta}}}$ aiming to let the models fail to generate meaningful images corresponding to $\mathcal{C}_f$:

$$\max_{\hat{\boldsymbol{\theta}}} \mathbb{E}_{\mathbf{x}_0 \sim \mathcal{D}_f}[-\log p_{\hat{\boldsymbol{\theta}}}(\mathbf{x}_0)] \propto \max_{\hat{\boldsymbol{\theta}}} \mathbb{E}_{\mathbf{x}_0 \sim \mathcal{D}_f} \left[ \sum_{t=2}^{T} \underbrace{\mathrm{KL}(q(\mathbf{x}_{t-1}|\mathbf{x}_t,\mathbf{x}_0)\|p_{\hat{\boldsymbol{\theta}}}(\mathbf{x}_{t-1}|\mathbf{x}_t))}_{\mathcal{L}_{t-1}(\hat{\boldsymbol{\theta}},\mathcal{D}_f)} \right]. \quad (3)$$

Given $\mathbf{x}_0 \sim \mathcal{D}_f$, the ground-truth backward distribution $q(\mathbf{x}_{t-1}|\mathbf{x}_t, \mathbf{x}_0) = \mathcal{N}(\mathbf{x}_{t-1}; \tilde{\boldsymbol{\mu}}_t, \tilde{\beta}_t^2 \mathbf{I}_d)$ guides $\mathbf{x}_T \sim \mathcal{N}(\mathbf{0}, \mathbf{I}_d)$ or $\mathbf{x}_t = \sqrt{\bar{\alpha}_t}\mathbf{x}_0 + \sqrt{1-\bar{\alpha}_t}\boldsymbol{\epsilon}$ with $\boldsymbol{\epsilon} \sim \mathcal{N}(\mathbf{0}, \mathbf{I}_d)$ to get back the forgetting data example $\mathbf{x}_0$ for obtaining meaningful examples. Additionally, the learnable backward distribution $p_{\boldsymbol{\theta}}(\mathbf{x}_{t-1}|\mathbf{x}_t)$ aims to mimic the ground-truth backward distribution $q(\mathbf{x}_{t-1}|\mathbf{x}_t, \mathbf{x}_0) = \mathcal{N}(\mathbf{x}_{t-1}; \tilde{\boldsymbol{\mu}}_t, \tilde{\beta}_t^2 \mathbf{I}_d)$ by minimizing the KL divergence $\mathrm{KL}(q(\mathbf{x}_{t-1}|\mathbf{x}_t, \mathbf{x}_0)\|p_{\hat{\boldsymbol{\theta}}}(\mathbf{x}_{t-1}|\mathbf{x}_t))$ for earning good trajectories that can reach the forgetting data example $\mathbf{x}_0$ proximally. To deviate $\mathbf{x}_T \sim \mathcal{N}(\mathbf{0}, \mathbf{I}_d)$ or $\mathbf{x}_t = \sqrt{\bar{\alpha}_t}\mathbf{x}_0 + \sqrt{1-\bar{\alpha}_t}\boldsymbol{\epsilon}$ with $\boldsymbol{\epsilon} \sim \mathcal{N}(\mathbf{0}, \mathbf{I}_d)$ from these trajectories, take DDPM as an example, we replace $q(\mathbf{x}_{t-1}|\mathbf{x}_t, \mathbf{x}_0) = \mathcal{N}(\mathbf{x}_{t-1}; \tilde{\boldsymbol{\mu}}_t, \tilde{\beta}_t^2 \mathbf{I}_d)$ where $\tilde{\boldsymbol{\mu}}_t = \frac{1}{\sqrt{\alpha_t}}(\mathbf{x}_t - \frac{\beta_t}{\sqrt{1-\bar{\alpha}_t}}\boldsymbol{\epsilon}_t)$ with $\boldsymbol{\epsilon} \sim \mathcal{N}(\mathbf{0}, \mathbf{I}_d)$ by $\hat{q}(\mathbf{x}_{t-1}|\mathbf{x}_t, \mathbf{x}_0) = \mathcal{N}(\mathbf{x}_{t-1}; \hat{\boldsymbol{\mu}}_t, \tilde{\beta}_t^2 \mathbf{I}_d)$ where $\hat{\boldsymbol{\mu}}_t = \frac{1}{\sqrt{\alpha_t}}(\mathbf{x}_t - \frac{\beta_t}{\sqrt{1-\bar{\alpha}_t}}\hat{\boldsymbol{\epsilon}}_t)$ with $\hat{\boldsymbol{\epsilon}}_t \sim \mathcal{U}(\mathbf{0}, \boldsymbol{I})$.

**Remark.** *$\hat{\boldsymbol{\epsilon}}_t$ could be any distribution different from $\boldsymbol{\epsilon}_t$, we choose uniform distribution $\hat{\boldsymbol{\epsilon}}_t \sim \mathcal{U}(\mathbf{0}, \boldsymbol{I})$ for experiments due to no extra hyper-parameters being needed. Appendix A.2 also show results for $\hat{\boldsymbol{\epsilon}}_t \sim \mathcal{N}(\boldsymbol{\mu}, \mathbf{I}_d)$ where $\boldsymbol{\mu} \neq \mathbf{0}$.*

Then, we define the following objective function

$$f(\hat{\boldsymbol{\theta}}, \mathcal{D}_f) := \mathbb{E}_{\mathbf{x}_0 \sim \mathcal{D}_f, \boldsymbol{\epsilon} \sim \mathcal{N}(\mathbf{0}, \boldsymbol{I}), \hat{\boldsymbol{\epsilon}} \sim \mathcal{U}(\mathbf{0}, \mathbf{1})} \left[ a \cdot \left\| \hat{\boldsymbol{\epsilon}} - \boldsymbol{\epsilon}_{\hat{\boldsymbol{\theta}}}(\sqrt{\bar{\alpha}_t}\mathbf{x}_0 + \sqrt{1-\bar{\alpha}_t}\boldsymbol{\epsilon}, t) \right\|^2 \right]. \quad (4)$$

With this, the scrubbed model $\boldsymbol{\epsilon}_{\hat{\boldsymbol{\theta}}}$ would tend to predict $\hat{\boldsymbol{\epsilon}}$ given $\mathbf{x}_t$. As such, for the forgetting data, the approximator $\boldsymbol{\epsilon}_{\hat{\boldsymbol{\theta}}}$ cannot learn the correct denoising distribution and thus cannot help to generate corresponding images when sampling.

---

**Algorithm 1** *EraseDiff*.

---

**Input:** Well-trained model $\epsilon_{\boldsymbol{\theta}_0}$, forgetting data $\mathcal{D}_f$ and subset of remaining data $\mathcal{D}_{rs} \subset \mathcal{D}_r$, outer iteration number $S$ and inner iteration number $K$, learning rate $\zeta$ and hyparameter $\lambda$.
**Output:** Parameters $\boldsymbol{\theta}^*$ for the scrubbed model.
  1: **for** iteration $s$ in $S$ **do**
  2:     $\phi_s^0 = \boldsymbol{\theta}_s$.
  3:     Get $\phi_s^K$ by $K$ steps of gradient descent on $f(\phi_s, \mathcal{D}_f)$ start from $\phi_s^0$ using Eq. (8).
  4:     Set $\hat{f}(\phi_s, \mathcal{D}_f) = f(\phi_s, \mathcal{D}_f) - f(\phi_s^K, \mathcal{D}_f)$.
  5:     Update the model: $\boldsymbol{\theta}_{s+1} = \boldsymbol{\theta}_s - \zeta(\nabla_{\boldsymbol{\theta}_s}\mathcal{F}(\boldsymbol{\theta}_s, \mathcal{D}_{rs}) + \lambda\nabla_{\phi_s}\hat{f}(\phi_s, \mathcal{D}_f))$.
  6: **end for**

---

**Final objective.** We strive to learn a model $\epsilon_{\boldsymbol{\theta}}$, such that when we apply the unlearning algorithm $\text{Alg}(\boldsymbol{\theta}, \mathcal{D}_f)$ to erase the influence of the forgetting data $\mathcal{D}_f$ in the model, the resulting model with parameters $\boldsymbol{\theta}^*$ should still be adept at generating samples having the same distribution as $\mathcal{D}_r$. To achieve this, given a well-trained diffusion model with parameters $\boldsymbol{\theta}_0$ over the data $\mathcal{D}$, for $t = 2, \cdots, T$, refer to Rajeswaran et al. (2019), we update the model with the following objective

$$\boldsymbol{\theta}^* := \arg\min_{\boldsymbol{\theta}} \mathcal{F}(\boldsymbol{\theta}), \quad \text{where } \mathcal{F}(\boldsymbol{\theta}) = \mathcal{L}(\text{Alg}(\boldsymbol{\theta}, \mathcal{D}_f), \mathcal{D}_r), \tag{5}$$

where $\mathbb{R}^d \ni \phi^*(\boldsymbol{\theta}) := \text{Alg}(\boldsymbol{\theta}, \mathcal{D}_f) = \arg\min_{\phi} f(\phi, \mathcal{D}_f)$, $\phi$ is initialized with $\boldsymbol{\theta}$ and $\boldsymbol{\theta}$ starts from $\boldsymbol{\theta}_0$. Starting from the parameters $\boldsymbol{\theta}_0$, for incoming forgetting data $\mathcal{D}_f$, Eq. (5) updates the approximator $\epsilon_{\boldsymbol{\theta}_0}$ to scrub the information about $\mathcal{D}_f$, and the updated weights $\boldsymbol{\theta}^*$ can also preserve the model utility over the remaining data $\mathcal{D}_r$. To encompass the unconditional and conditional image generation scenarios, Eq. (5) can be further rewritten as

$$\min_{\boldsymbol{\theta}} \mathbb{E}_{\mathbf{x}_0 \sim \mathcal{D}_r, c \sim \mathcal{C}_r}\left[a \cdot \left\|\boldsymbol{\epsilon} - \epsilon_{\boldsymbol{\theta}}(\sqrt{\bar{\alpha}_t}\mathbf{x}_0 + \sqrt{1 - \bar{\alpha}_t}\boldsymbol{\epsilon}, t, c)\right\|^2\right],$$

$$\text{s.t. } \boldsymbol{\theta} \in \arg\min_{\hat{\boldsymbol{\theta}}} \mathbb{E}_{\mathbf{x}_0 \sim \mathcal{D}_f, \hat{\boldsymbol{\epsilon}} \sim \mathcal{U}(\mathbf{0},\mathbf{1}), c \sim \mathcal{C}_f}\left[a \cdot \left\|\hat{\boldsymbol{\epsilon}} - \epsilon_{\hat{\boldsymbol{\theta}}}(\sqrt{\bar{\alpha}_t}\mathbf{x}_0 + \sqrt{1 - \bar{\alpha}_t}\boldsymbol{\epsilon}, t, c)\right\|^2\right], \tag{6}$$

where $c$ would be $\emptyset$ for unconditional diffusion models and labels for conditional diffusion models. Eqs. (5) and (6) aim to keep the model utility over the remaining data $\mathcal{D}_r$, but misguide it $\mathcal{D}_f$ by deviating the learnable reverse process from the ground-truth denoising procedure.

## 3.2 SOLUTION.

Eq. (5) could be viewed as a bilevel optimization (BO) problem, where the goal is to minimize an outer objective $\mathcal{F}(\boldsymbol{\theta}, \mathcal{D}_r)$ whose variables include the solution of another minimization problem w.r.t. an inner objective $f(\phi, \mathcal{D}_f)$. BO problem is challenging, and most existing methods for BO require expensive manipulation of the Hessian matrix (Liu et al., 2022b). We adopt a practical and efficient algorithm that depends only on first-order gradient information proposed by Liu et al. (2022a) to solve Eq. (5). Our method is shown in Algorithm 1 (Detailed version can be found in Algorithm 2 in Appendix A.1). The objective shown in Eq. (5) will be reformulated as a single-level constrained optimization problem

$$\min_{\boldsymbol{\theta}} \mathcal{F}(\boldsymbol{\theta}, \mathcal{D}_r), \quad \text{s.t. } \hat{f}(\phi, \mathcal{D}_f) := f(\phi, \mathcal{D}_f) - f(\phi^*(\boldsymbol{\theta}), \mathcal{D}_f) \leq 0, \tag{7}$$

where $\phi^*(\boldsymbol{\theta}) := \text{Alg}(\boldsymbol{\theta}, \mathcal{D}_f) = \arg\min_{\phi} f(\phi, \mathcal{D}_f)$ and Eq. (7) yields first-order algorithms for non-convex functions. Specifically, $\phi^*(\boldsymbol{\theta})$ is approximated by running $K$ steps of gradient descent of $f(\phi, \mathcal{D}_f)$ over $\mathcal{D}_f$, we set $\phi^0 = \boldsymbol{\theta}$, so

$$\phi^k \equiv \text{Alg}(\boldsymbol{\theta}, \mathcal{D}_f) = \phi^{k-1} - \zeta_f \nabla_{\phi^{k-1}} f(\phi^{k-1}, \mathcal{D}_f), k = 1, \ldots, K, \tag{8}$$

where $\zeta_f = \zeta$ is the learning rate. We could obtain the approximation $\hat{f}(\phi, \mathcal{D}_f) = f(\phi, \mathcal{D}_f) - f(\phi^K, \mathcal{D}_f)$. Then, with $\phi = \boldsymbol{\theta}$ and the subset $\mathcal{D}_{rs} \subset \mathcal{D}_r$, the model will be updated via

$$\boldsymbol{\theta}^* = \boldsymbol{\theta} - \zeta(\nabla_{\boldsymbol{\theta}}\mathcal{F}(\boldsymbol{\theta}, \mathcal{D}_{rs}) + \lambda\nabla_{\phi}\hat{f}(\phi, \mathcal{D}_f)). \tag{9}$$

**Remark.** *We use $\lambda = 0.1$ by default. $\lambda$ could also be automatically computed for different iterations as shown in the study (Liu et al., 2022a) and results can be found in Appendix A.2.*

## 4 RELATED WORK

**Memorization in generative models.** Privacy of generative models has been studied a lot for GANs (Feng et al., 2021; Meehan et al., 2020; Webster et al., 2021) and generative language models (Carlini et al., 2022; 2021; Jagielski et al., 2022; Tirumala et al., 2022; Lee et al., 2023). These generative models often risk replication from their training data. Recently, several studies (Carlini et al., 2023; Somepalli et al., 2023b;a; Vyas et al., 2023) investigated these data replication behaviors in diffusion models, raising concerns about the privacy and copyright issues. Possible mitigation strategies are deduplicating training data and randomizing conditional information (Somepalli et al., 2023b;a), or training models with differential privacy (DP) (Abadi et al., 2016; Dwork et al., 2006; Dwork, 2008; Dockhorn et al., 2022). However, Carlini et al. (2023) show that deduplication is not a perfect solution, and leveraging DP-SGD (Abadi et al., 2016) may cause the training to diverge.

**Malicious misuse.** Diffusion models usually use training data from varied open sources and when such unfiltered data is employed, there is a risk of it being tainted(Chen et al., 2023b) or manipulated (Rando et al., 2022), resulting in inappropriate generation (Schramowski et al., 2023). They also risk the imitation of copyrighted content, such as mimicking the artistic style (Gandikota et al., 2023a; Shan et al., 2023). To counter inappropriate generation, data censoring (Gandhi et al., 2020; Birhane & Prabhu, 2021; Nichol et al., 2021; Schramowski et al., 2022) where excluding black-listed images before training, and safety guidance where diffusion models will be updated away from the inappropriate/undesired concept (Gandikota et al., 2023a; Schramowski et al., 2023) are proposed. Shan et al. (2023) propose protecting artistic style by adding barely perceptible perturbations into the artworks before public release. Yet, Rando et al. (2022) argue that filtering can still generate disturbing content that bypasses the filter. Chen et al. (2023b) highlight the susceptibility of diffusion models to poison attacks, where target images will be generated with specific triggers.

**Machine unlearning.** Removing data directly involves retraining the model from scratch, which is inefficient and impractical. Thus, to reduce the computational overhead, efficient machines unlearning methods Romero et al. (2007); Karasuyama & Takeuchi (2010); Cao & Yang (2015); Ginart et al. (2019); Bourtoule et al. (2021); Wu et al. (2020); Guo et al. (2020); Golatkar et al. (2020); Mehta et al. (2022); Sekhari et al. (2021); Chen et al. (2023a); Tarun et al. (2023b) have been proposed. Gandikota et al. (2023a); Heng & Soh (2023); Gandikota et al. (2023b) recently introduce unlearning in diffusion models. Gandikota et al. (2023a;b) mainly focuses on text-to-image models and high-level visual concept erasure. Heng & Soh (2023) adopt Elastic Weight Consolidation (EWC) and Generative Replay (GR) from continual learning to perform unlearning effectively without access to the training data. In this work, we present an unlearning algorithm for diffusion-based generative models. Specifically, we update the model parameters where the outer objective is to maintain utility for the remaining data $\mathcal{D}_r$, and the inner objective is to erase information about the forgetting data $\mathcal{D}_f$. Note that both the inner and outer objectives in our formulation and MAML (Rajeswaran et al., 2019) are directed at optimizing the model parameters. Our formulation distinction from MAML lies in its focus: we are not seeking a model adaptable to unlearning but one that effectively erases the influence of data points on the model. Our method allows for easily equipping any plug-and-play loss function(e.g., (Ruiz et al., 2023; Song et al., 2023)) into the unlearning procedure. This is as easy as changing the outer loop to incorporate plug-and-play loss functions if they are deemed to improve the quality of generation.

## 5 EXPERIMENT

We evaluate the proposed unlearning method *EraseDiff* in various scenarios, including removing images with specific classes/races/attributes, to answer the following research questions (RQs):

**RQ1:** Is the proposed method able to remove the influence of the forgetting data in the diffusion models?

**RQ2:** Is the proposed method able to preserve the model utility while removing the forgetting data?

**RQ3:** Is the proposed method efficient in removing the data?

**RQ4:** Can typical machine unlearning methods be applied to diffusion models, and how does the proposed method compare with these unlearning methods?

**RQ5:** How does the proposed method perform on the public well-trained models from Hugging Face[2]?

---

[2]https://huggingface.co/models

Table 1: FID score over forgetting classes on conditional DDIM. FID score is evaluated on 50K generated images (5K per class for CIFAR10, 12.5K per class for UTKFace). 'Accuracy': the classification accuracy on the corresponding generated images. The pre-trained classifiers have a classification accuracy of 0.81 on CIFAR10 and 0.84 on UTKFace. The generated images conditioned on the forgetting classes significantly deviate from the corresponding real counterparts.

| | CIFAR10 | | | UTKFace | |
|---|---|---|---|---|---|
| | $c = 2 \uparrow$ | $c = 8 \uparrow$ | Accuracy$\downarrow$ | $c = 3 \uparrow$ | Accuracy$\downarrow$ |
| Unscrubbed | 19.62 | 12.05 | 0.8622 | 8.87 | 0.7614 |
| *EraseDiff* (Ours) | **256.27** (+236.65) | **294.08** (+282.03) | **0.0026** | **330.33** (+321.46) | **0.0000** |

Figure 1: Distribution of loss value on the forgetting data $\mathcal{D}_f$ and unseen data $\mathcal{D}_t$. Models scrubbed by our unlearning algorithm have lower MIA accuracy than unscrubbed models, indicating that our methodology successfully scrubbed the data influence.

## 5.1 SETUP

Experiments are reported on CIFAR10 (Krizhevsky et al., 2009), UTKFace (Zhang et al., 2017), CelebA (Liu et al., 2015), and CelebA-HQ (Lee et al., 2020) datasets. By default, our scrubbed models use $\sim 8K$ images from $\mathcal{D}_r$ during the unlearning process. Implementation details and more results (e.g., concept/attribute unlearning on CelebA and CelebA-HQ, class unlearning on CIFAR-10 and UTKFace), including the ablation study (ie., the hyper-parameter $\lambda$ that control the balance between $\mathcal{D}_r$ and $\mathcal{D}_f$, the number of images from $\overline{\mathcal{D}_r}$), can be found in the Appendix.

**Baselines.** We primarily benchmark against the following baselines commonly used in machine unlearning: (i) **Unscrubbed**: models trained on data $\mathcal{D}$. Unlearning algorithms should scrub information from its parameters. (ii) **Retrain**: models obtained by retraining from scratch on the remaining data $\mathcal{D}_r$. (iii) **Finetune** (Golatkar et al., 2020): finetuning models on the remaining data $\mathcal{D}_r$, ie., catastrophic forgetting. (iv) **NegGrad** (Golatkar et al., 2020): gradient ascent on the forgetting data $\mathcal{D}_f$. (v) **BlindSpot** (Tarun et al., 2023b): the state-of-the-art unlearning algorithm for regression. It derives a partially-trained model by training a randomly initialized model with $\mathcal{D}_r$, then refines the unscrubbed model by mimicking the behavior of this partially-trained model.

**Metrics.** Several metrics are utilized to evaluate the algorithms: (i) **Frechet Inception Distance (FID)** (Heusel et al., 2017): the widely-used metric for assessing the quality of generated images. (ii) **Accuracy**: the classification rate of a pre-trained classifier, with a ResNet architecture (He et al., 2016) used to classify generated images conditioned on the forgetting classes. A lower classification value indicates superior unlearning performance. (iii) **Membership inference attack (MIA)**: a standard metric for verifying unlearning effect in classification tasks. However, due to the high-quality image generation of diffusion models, which closely resemble real images, MIA might not distinctly differentiate between training and unseen images. (iv) **Kullback–Leibler (KL)** divergence: distance between the approximator's output $\epsilon_T$ distribution and the standard Gaussian noise $\epsilon$ distribution. (v) **Weight Distance (WD)** (Tarun et al., 2023a): distance between the Retrain models' weights and other scrubbed models' weights. WD gives additional insights about the amount of information remaining in the models about the forgetting data.

## 5.2 EFFECTIVENESS OF DATA REMOVAL

We aim to unlearn images conditioned on classes of birds and ships (represented by $c = 2$ and $c = 8$) on CIFAR10, faces with the last ethnicity labeled as Indian ($c = 3$) on UTKFace, and images with attribute blond hair (unconditional case) on CelebA and CelebA-HQ, respectively. Results in different scenarios can be found in Appendix A.2. For an effective unlearning algorithm, the

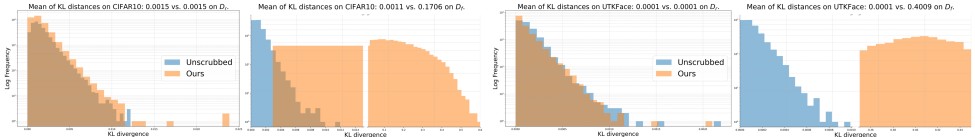

(a) CIFAR10        (b) UTKFace

Figure 2: Images generated by conditional DDIM. Images in the green dashed box are generated by conditioning on the remaining labels $\mathcal{C}_r$ and those in the red solid box are generated by conditioning on the forgetting classes $\mathcal{C}_f$. Our unlearning algorithm could successfully scrub the information of $\mathcal{C}_f$ carried by the models while maintaining the model utility over $\mathcal{C}_r$.

Figure 3: KL distance between the approximator output $\epsilon_T$ and standard Gaussian noise $\epsilon$. Our unlearning algorithm maintains the model utility on $\mathcal{D}_r$ while ensuring that the predicted distribution significantly diverges from the predefined standard Gaussian noise distribution for $\mathcal{D}_f$.

scrubbed models are expected to contain little information about the forgetting data. In this section, we first show the effectiveness of our proposed unlearning algorithm on diffusion models.

**FID score.** Tab. 1 presents the results of FID on the generated images conditioned on the forgetting classes $\mathcal{C}_f$. FID scores are computed between the generated images and the corresponding images with the same labels from the training data $\mathcal{D}_f$. As shown in Tab. 1, generated images conditioned on $\mathcal{C}_f$ from the models scrubbed by our proposed unlearning algorithm have larger FID scores than those from the unscrubbed models, indicating that our algorithm successfully erases the information of the forgetting data carried by the models. For example, on CIFAR10, the FID score of generated images conditioned on the label birds increased from 19.62 to 256.27, indicating that these generated bird images are notably dissimilar from the bird images present in the training data.

**Accuracy.** The classification accuracy of the generated images conditioned on $\mathcal{D}_f$ decreased from around 0.86 to 0.002 on CIFAR10, and from 0.76 to 0 on UTKFace, respectively. This demonstrates that, among the generated 10K images on CIFAR10 and 25K images on UTKFace, only 26 images are classified into the categories of birds or ships on CIFAR10 and no faces are classified as Indian celebrities, further verifying the effectiveness of our unlearning algorithm.

**Loss distribution.** To further measure the effectiveness of data removal, we report the results of MIA accuracy to indicate whether the data is used to train the models. The attack model is a binary classifier and is trained using the loss value w.r.t. the forgetting data, and data from the test set. As shown in Fig. 1, after scrubbing the models using our proposed unlearning algorithm *EraseDiff*, the distribution of loss values on $\mathcal{D}_f$ is similar to that on the test data, indicating that our methodology successfully erased the information of the forgetting data carried by the model.

Ideally, the loss values w.r.t. the training data would be lower than those w.r.t. the data from the test set. However, diffusion models have impressive capacities to generate diverse and high-quality images, leading to difficulty in distinguishing between the generated images and the real images in the training data. For example, on CIFAR10 shown in Fig. 1, the distribution of loss values on $\mathcal{D}_f$ is quite similar to that on the unseen data for the unscrubbed model, resulting in the instability of MIA performance. Advanced methods to measure MIA could be leveraged, we leave it for future studies.

**KL divergence.** Here, we propose computing the KL divergence distance between the model output distribution and the pre-defined Gaussian noise distribution. The diffusion process would add noise to the clean input $\mathbf{x}$, consequently, the model output $\epsilon_T$ distribution given these noisy images $\mathbf{x}_t$ tends to converge towards the Gaussian noise distribution $\epsilon$. As depicted in Fig. 3, the y-axis represents the number of occurrences of the specific distance value. We can see that the distance between the output distribution w.r.t. $\mathcal{C}_r$ and the Gaussian distribution for our scrubbed model is close to that of the unscrubbed model, while the distance between the output distribution w.r.t. the $\mathcal{C}_f$ is further away from the Gaussian distribution for our scrubbed model than the unscrubbed model. This means for $\mathcal{D}_r$, our scrubbed models exhibit behavior akin to the unscrubbed models, while for $\mathcal{D}_f$, our scrubbed models would generate images significantly deviate from $\epsilon$. These results under-

Table 2: FID score over the remaining classes on conditional DDIM. FID score is evaluated on 50K generated images (5K per class for CIFAR10, 12.5K per class for UTKFace). The quality of the generated image conditioned on the remaining classes drops a little after scrubbing (examples in Fig. 2) but it is acceptable given the significant unlearning impact of the forgetting classes.

| | CIFAR10 | | | | | | | | UTKFace | | |
|---|---|---|---|---|---|---|---|---|---|---|---|
| | $c=0\downarrow$ | $c=1\downarrow$ | $c=3\downarrow$ | $c=4\downarrow$ | $c=5\downarrow$ | $c=6\downarrow$ | $c=7\downarrow$ | $c=9\downarrow$ | $c=0\downarrow$ | $c=1\downarrow$ | $c=2\downarrow$ |
| Unscrubbed | 17.04 | 9.67 | 19.88 | 14.78 | 20.56 | 17.16 | 11.53 | 11.44 | 7.37 | 11.28 | 9.72 |
| *EraseDiff* (Ours) | 29.61 | 22.10 | 28.65 | 27.68 | 35.59 | 23.93 | 21.24 | 24.85 | 8.08 | 13.52 | 12.37 |

Table 3: Results on CIFAR10 trained with conditional DDIM. Although NegGrad and BlindSpot could scrub the information of the forgetting classes, they cannot maintain the model utility. Other methods fail to scrub the relevant information completely. Examples are shown in Fig. 4.

| Method | FID over forgetting classes | | Accuracy↓ | FID over remaining classes | | | | | | | | WD↓ |
|---|---|---|---|---|---|---|---|---|---|---|---|---|
| | $c=2\uparrow$ | $c=8\uparrow$ | | $c=0\downarrow$ | $c=1\downarrow$ | $c=3\downarrow$ | $c=4\downarrow$ | $c=5\downarrow$ | $c=6\downarrow$ | $c=7\downarrow$ | $c=9\downarrow$ | |
| Unscrubbed | 19.62 | 12.05 | - | 17.04 | 9.67 | 19.88 | 14.78 | 20.56 | 17.16 | 11.53 | 11.44 | - |
| Retrain | 152.39 | 139.62 | 0.0135 | 17.39 | 9.57 | 20.05 | 14.65 | 20.19 | 17.85 | 11.63 | 10.85 | 0.0000 |
| Finetune | 31.64 | 21.22 | 0.7001 | **20.49** | **12.38** | **23.47** | **17.80** | **25.51** | **18.23** | **14.43** | **16.09** | 1.3616 |
| NegGrad | 322.67 | 229.08 | 0.4358 | 285.25 | 290.57 | 338.49 | 290.23 | 312.44 | 339.43 | 320.63 | 278.03 | **1.3533** |
| BlindSpot | **349.60** | **335.69** | 0.1167 | 228.92 | 181.88 | 288.88 | 252.42 | 242.16 | 278.62 | 192.67 | 195.27 | 1.3670 |
| *EraseDiff* (Ours) | 256.27 | 294.08 | **0.0026** | 29.61 | 22.10 | 28.65 | 27.68 | 35.59 | 23.93 | 21.24 | 24.85 | 1.3534 |

score the efficacy of our unlearning algorithm in scrubbing the unscrubbed model concerning the information of $\mathcal{D}_f$, all the while preserving the model utility for $\mathcal{D}_r$.

**Visualizations.** To provide a more tangible assessment of unlearning effectiveness, we visualize some generated images. Fig. 2 showcases samples generated by the conditional DDIM models from Tab. 1. When conditioned on $\mathcal{D}_f$, the scrubbed models predominantly produce unrecognizable images. More examples and examples of the unconditional case can be found in Appendix A.2.

### 5.3 EFFICACY OF PRESERVING MODEL UTILITY.

For an effective unlearning algorithm, the scrubbed models are also expected to maintain utility over the remaining classes. So we present the FID scores over the generated images conditioned on $\mathcal{C}_r$ from the scrubbed models in Tab. 2. The FID scores increase compared with the generated images from the original models. As shown in Fig. 2, it's evident that the quality of generated images experiences a slight decrease. However, given the substantial unlearning impact of the forgetting classes, this drop in quality is acceptable. Furthermore, as shown in Figs. 1 and 3, our scrubbed models could obtain similar results as the unscrubbed models over the remaining classes.

### 5.4 EFFICIENCY OF THE ALGORITHM

Assuming that the computational complexity for a single epoch over the data $\mathcal{D}$ is represented as $\mathcal{O}(h(N))$, similarly, a single epoch of training over the remaining data $\mathcal{D}_r$ is denoted as $\mathcal{O}(h(N_r))$, and a single epoch of training over the forgetting data $\mathcal{D}_f$ is denoted as $\mathcal{O}(h(N-N_r)) = \mathcal{O}(h(N_f))$. Then, the computational complexity for training the unscrubbed models would be $\mathcal{O}(E_1 \cdot h(N))$ for $E_1$ epochs, and the computational complexity for training the retrained models would be $\mathcal{O}(E_2 \cdot h(N_r))$ for $E_2(E_2 < E_1)$ epochs. Our unlearning algorithm exhibits computational complexity as $\mathcal{O}(E \cdot S(K \cdot h(N_f) + 2h(N_{rs})))$ for $E$ epochs, where $(ES(K+2)) \ll E_2, N_{rs} \ll N_r$ and $N_f \ll N_r$. Consider the experiments on CIFAR10, as detailed in Tab. 3. On A100, it takes over 32 hours for unscrubbed models and roughly 27 hours for the retrained models to complete their training. In contrast, our algorithm efficiently scrubs models in just around 10 minutes.

### 5.5 COMPARISON WITH OTHER ALGORITHMS

We now evaluate the proposed unlearning algorithm *EraseDiff* in comparison with other machine unlearning methods. As illustrated in Tab. 3, the NegGrad model and BlindSpot model can erase the information related to forgetting classes $\mathcal{C}_f$, but struggles to retain the model utility. For example, images generated conditioned on $\mathcal{C}_f$ (birds and ships) yield the FID score exceeding 200, and ~2000

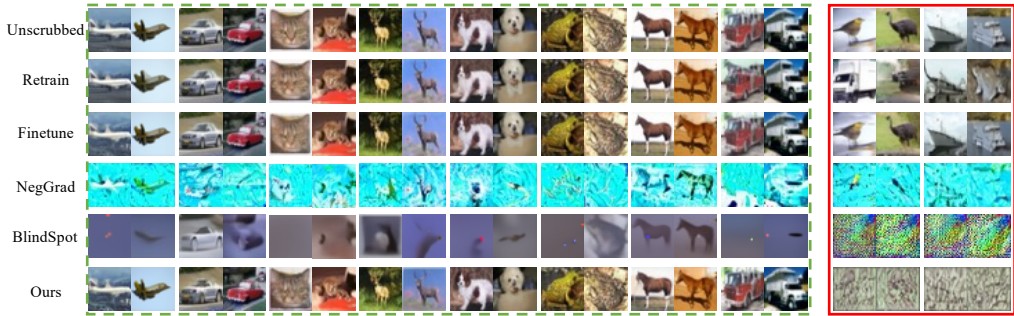

Figure 4: Images generated by conditional DDIM. Images in the green dashed box are generated by conditioning on the remaining labels $\mathcal{C}_r$ and those in the red solid box are generated by conditioning on the forgetting classes $\mathcal{C}_f$. 'Retrain' generates images randomly for $\mathcal{C}_f$. 'NegGrad' and 'BlindSpot' successfully scrub the information regarding $\mathcal{C}_f$ as well but cannot withhold the model utility. Ours can sufficiently erase relevant information and maintain the model utility.

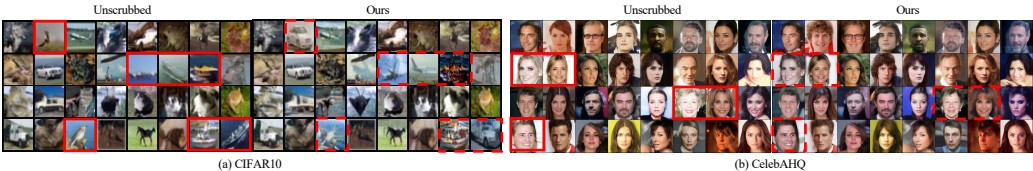

Figure 5: Images generated by the well-trained unconditional DDPM from Hugging Face and our corresponding scrubbed models. Images in the red solid box have the label belonging to the forgetting class/attribute, those in the red dashed box are the corresponding images generated by ours.

of these generated images are classified into the birds or ships categories. However, for the generated images conditioned on the remaining classes $\mathcal{C}_r$, the FID scores also surpass 200, as visualized in Fig. 4, these generated images appear corrupted. As shown in Tab. 3 and Fig. 4, other baseline models can preserve the model utility over $\mathcal{C}_r$ but fall short in sufficiently erasing the information regarding $\mathcal{C}_f$. Our proposed unlearning algorithm, instead, adeptly scrubs pertinent information while upholding model utility. Furthermore, we can observe that our scrubbed model has a WD of 1.3534, while the minimal WD is 1.3533, indicating that our scrubbed models are in close alignment with the retrained model where the forgetting data $\mathcal{D}_f$ never attends in the training process.

## 5.6 USER STUDY

Subsequently, we assess the proposed unlearning method *EraseDiff* on the proficiently trained unconditional DDPM models on CIFAR10 and CelebA-HQ from Hugging Face. Fig. 5 presents examples of generated images from the original models and our corresponding scrubbed models. We can observe that when the unscrubbed models generate images whose categories can be the forgetting classes, the scrubbed models consistently generate images that either belong exclusively to $\mathcal{C}_r$ or are unrecognizable, further indicating the effectiveness of the proposed unlearning algorithm.

## 6 CONCLUSION AND DISCUSSION

In this work, we first explored the unlearning problem in diffusion models and proposed an effective unlearning method. Comprehensive experiments on unconditional and conditional diffusion models demonstrate the proposed algorithm's effectiveness in data removal, its efficacy in preserving the model utility, and its efficiency in unlearning. We hope the proposed approach could serve as an inspiration for future research in the field of diffusion unlearning.

However, our present evaluations focus on categories or subsets with significant common characteristics, potentially overlooking a broader range of problem scenarios. Future directions for diffusion unlearning could include assessing fairness post-unlearning, implementing weight unlearning to prioritize different samples, using advanced privacy-preserving training techniques, developing effective multi-task frameworks, and developing distinct optimization strategies.

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

# A APPENDIX

## A.1 SOCIAL IMPACT

Diffusion models have experienced rapid advancements and have shown the merits of generating high-quality data. However, concerns have arisen due to their ability to memorize training data and generate inappropriate content, thereby negatively affecting the user experience and society as a whole. Machine unlearning emerges as a valuable tool for correcting the algorithms and enhancing user trust in the respective platforms. It demonstrates a commitment to responsible AI and the welfare of its user base. However, while unlearning protects privacy, it may also hinder the ability of relevant systems and potentially lead to biased outcomes.

## A.2 IMPLEMENTATION DETAILS

Four and five feature map resolutions are adopted for CIFAR10 where image resolution is $32 \times 32$, UTKFace and CelebA where image resolution is scaled to $64 \times 64$, respectively. Our $32 \times 32$ model, and $64 \times 64$ model have around 36 million, and 79 million parameters, respectively. The well-trained unconditional DDPM models on CIFAR10[3] and CelebA-HQ[4] are downloaded from Hugging Face. We used A40 and A100 for all experiments. All models apply the linear schedule for the diffusion process. We set the batch size $B = 512$, $B = 128$, $B = 64$, $B = 16$ for CIFAR10, UTKFace and CelebA, CelebA-HQ respectively. The linear schedule is set from $\beta_1 = 10^{-4}$ to $\beta_T = 0.02$, the inference time step for DDIM is set to be 100, the guidance scale $w = 0.1$, and the probability $p_{uncond} = 0.1$ for all models. For the Unscrubbed and Retrain models, the learning rate is $3 \times 10^{-4}$ for CIFAR10 and $2 \times 10^{-4}$ for other datasets. We train the CIFAR10 model for 2000 epochs, the UTKFace and CelebA models for 500 epochs. For Finetune models, the learning rate is $3 \times 10^{-4}$ for CIFAR10 and $2 \times 10^{-4}$ for other datasets, all the models are finetuned on the remaining data $\mathcal{D}_r$ for 100 epochs. For NegGrad models, the learning rate is $1 \times 10^{-6}$ and all the models are trained on the forgetting data $\mathcal{D}_f$ for 5 epochs. For BlindSpot models, the learning rate is $2 \times 10^{-4}$. The partially-trained model is trained for 100 epochs on the remaining data $\mathcal{D}_r$ and then the scrubbed model is trained for 100 epochs on the data $\mathcal{D}$. For our scrubbed models, $N_{rs} = |\mathcal{D}_{rs}| \approx 8K$, the learning rate is $1 \times 10^{-6}$ CelebA-HQ and $2 \times 10^{-4}$ for other datasets. Note that the components (Dhariwal & Nichol, 2021; Nichol & Dhariwal, 2021) for improving the model performance are not applied in this work.

---

**Algorithm 2** *EraseDiff*.

**Input:** Well-trained model $\epsilon_{\boldsymbol{\theta}_0}$, forgetting data $\mathcal{D}_f$ and subset of remaining data $\mathcal{D}_{rs} \subset \mathcal{D}_r$, outer iteration number $S$ and inner iteration number $K$, learning rate $\zeta$ and hyparameter $\lambda$.
**Output:** Parameters $\boldsymbol{\theta}^*$ for the scrubbed model.
1: **for** iteration s in S **do**
2:     $\phi_s^0 = \boldsymbol{\theta}_s$.
3:     Get $\phi_s^K$ by $K$ steps of gradient descent on $f(\phi_s, \mathcal{D}_f)$ start from $\phi_s^0$ using Eq. (8):
        Sample $\{\mathbf{x}_0, c\} \subset \mathcal{D}_f, t \sim \text{Uniform}(1, \cdots, T), \boldsymbol{\epsilon} \sim \mathcal{N}(\mathbf{0}, \mathbf{I}_d)$,
        Compute $\nabla_{\phi_s^k} \|\hat{\boldsymbol{\epsilon}} - \epsilon_{\phi_s^k}(\sqrt{\bar{\alpha}_t}\mathbf{x}_0 + \sqrt{1 - \bar{\alpha}_t}\boldsymbol{\epsilon}, t, c)\|$.
        Get the constant loss $\mathcal{L}_{cs} = \|\hat{\boldsymbol{\epsilon}} - \epsilon_{\phi_s^K}(\sqrt{\bar{\alpha}_t}\mathbf{x}_0 + \sqrt{1 - \bar{\alpha}_t}\boldsymbol{\epsilon}, t, c)\|$ if $k = K$.
4:     Set the approximation:
        Sample $\{\mathbf{x}_0, c\} \subset \mathcal{D}_f, t \sim \text{Uniform}(1, \cdots, T), \boldsymbol{\epsilon} \sim \mathcal{N}(\mathbf{0}, \mathbf{I}_d)$,
        Compute the loss $\mathcal{L}_f = \|\hat{\boldsymbol{\epsilon}} - \epsilon_{\boldsymbol{\theta}_s}(\sqrt{\bar{\alpha}_t}\mathbf{x}_0 + \sqrt{1 - \bar{\alpha}_t}\boldsymbol{\epsilon}, t, c)\| - \mathcal{L}_{cs}$.
5:     Update the model:
        Sample $\{\mathbf{x}_0, c\} \subset \mathcal{D}_{rs}, t \sim \text{Uniform}(1, \cdots, T), \boldsymbol{\epsilon} \sim \mathcal{N}(\mathbf{0}, \mathbf{I}_d)$,
        Compute the loss $\mathcal{L}_s = \|\boldsymbol{\epsilon} - \epsilon_{\boldsymbol{\theta}_s}(\sqrt{\bar{\alpha}_t}\mathbf{x}_0 + \sqrt{1 - \bar{\alpha}_t}\boldsymbol{\epsilon}, t, c)\| + \lambda \mathcal{L}_f$,
        Update $\boldsymbol{\theta}_{s+1} = \boldsymbol{\theta}_s - \zeta \nabla_{\boldsymbol{\theta}_s} \mathcal{L}_s$.
6: **end for**

---

[3]https://huggingface.co/google/ddpm-cifar10-32
[4]https://huggingface.co/google/ddpm-ema-celebahq-256

---

**Algorithm 3** BlindSpot Unlearning (Tarun et al., 2023b).

---

**Input:** A well-trained model $\epsilon$ with parameters $\boldsymbol{\theta}_0$, a randomly initialized blind model $\epsilon_{\psi}(\cdot)$, forgetting data $\mathcal{D}_f$, remaining data $\mathcal{D}_r$ and all training data $\mathcal{D} = \mathcal{D}_f \cup \mathcal{D}_r$. The learning rate $\zeta$, the number of epochs $E_r$ and $E_u$, hyper-parameter $\lambda$.

**Output:** Parameters $\boldsymbol{\theta}^*$ for the scrubbed model.

1: Initialization $\boldsymbol{\theta} = \boldsymbol{\theta}_0$.
2: **for** $1, 2, \ldots, E_r$ **do**
3:      train the blind model $\epsilon_{\psi}(\cdot)$ with the remaining data $\mathcal{D}_r$.
4: **end for**
5: **for** $1, 2, \ldots, E_u$ **do**
6:      **for** $(\mathbf{x}_i, c_i) \in \mathcal{D}$ **do**
7:          $l_f^i = 1$ if $(\mathbf{x}_i, c_i) \in \mathcal{D}_f$ else $l_f^i = 0$.
8:          $\boldsymbol{\epsilon}_t = \epsilon_{\boldsymbol{\theta}}(\mathbf{x}_i, t, c_i)$, where $t$ is the timestep and $t \in [1, T]$.
9:          $\mathcal{L}_r = \mathcal{L}(\boldsymbol{\epsilon}_t, \boldsymbol{\epsilon})$ and $\mathcal{L}_f = \mathcal{L}(\boldsymbol{\epsilon}_t, \epsilon_{\psi}(\mathbf{x}_i, t, c_i))$.
10:          $\mathcal{L}_a = \lambda \sum_{j=1}^{k} \|\text{act}_j^{\boldsymbol{\theta}} - \text{act}_j^{\boldsymbol{\psi}}\|$, where $\text{act}_j$ is the output of each block in the UNet.
11:          $\mathcal{L} = (1 - l_f^i)\mathcal{L}_r + l_f^i(\mathcal{L}_f + \mathcal{L}_a)$.
12:          $\boldsymbol{\theta} = \boldsymbol{\theta} - \zeta \frac{\partial \mathcal{L}}{\partial \boldsymbol{\theta}}$.
13:      **end for**
14: **end for**

---

### A.3 MORE RESULTS

In the following, we present the results of Ablation studies, results when replacing $\boldsymbol{\epsilon} \sim \mathcal{N}(\mathbf{0}, \mathbf{I}_d)$ with $\hat{\boldsymbol{\epsilon}}_t \sim \mathcal{N}(\mathbf{0.5}, \mathbf{I}_d)$ for Eq. (4), results when sampling from the uniform distribution $\hat{\boldsymbol{\epsilon}}_t \sim \mathcal{U}(\mathbf{0}, \mathbf{1})$, and results when trying to erase different classes/races/attributes under the conditional and unconditional scenarios. We include new comparisons (e.g., without access to $\mathcal{D}_r$, subjected to adversarial attacks, two alternative formulations to perform unlearning) in Tabs. 4 to 8 and Figs. 17 to 24. In general, with more remaining data during the unlearning process, the generated image quality over the remaining classes $\mathcal{C}_r$ would be better while those over the forgetting classes $\mathcal{C}_f$ would be worse. With generated images for unlearning, the image quality after scrubbing the model would be worse, but still surpasses other methods. When subjected to adversarial attacks, the quality of generated images of all models would decrease along with the step size of the attack increases, but the scrubbed model still would not contain information about $\mathcal{C}_f$. Simultaneously updating the model parameters can destroy the information about $\mathcal{C}_f$, but would also result in a significant drop in image quality $\mathcal{C}_r$. Disjoint optimization does not work as the second phase could bring back information about $\mathcal{C}_f$.

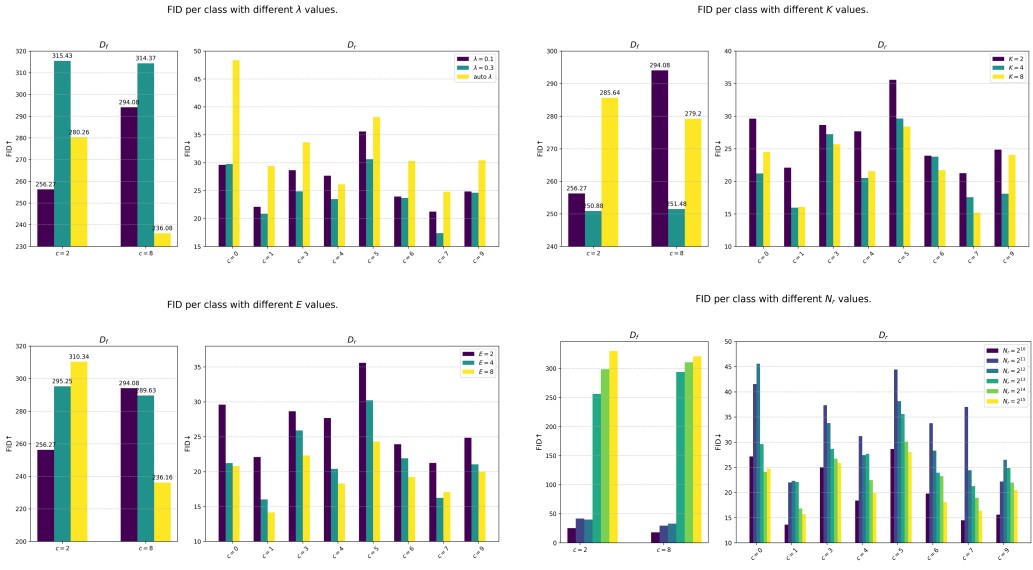

Figure 6: Ablation results with conditional DDIM on CIFAR10.

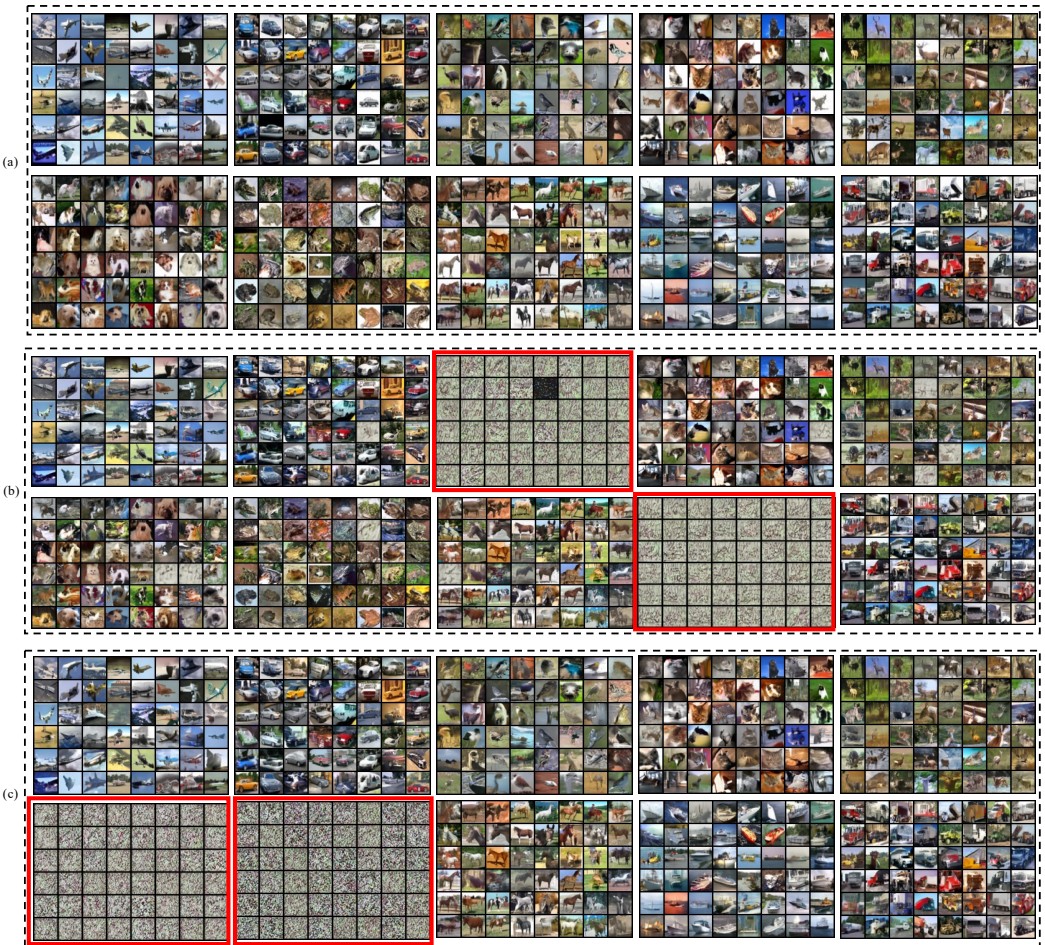

Figure 7: Conditional DDIM on CIFAR-10. (a) Generated images by the unscrubbed model. (b) and (c) are generated images by our scrubbed model when forgetting classes are $\mathcal{C}_f = \{c_2, c_8\}$, and $\mathcal{C}_f = \{c_5, c_6\}$, respectively. Images in the red solid box are generated by conditioning on the forgetting classes $\mathcal{C}_f$, others are generated by conditioning on the remaining classes $\mathcal{C}_r$.

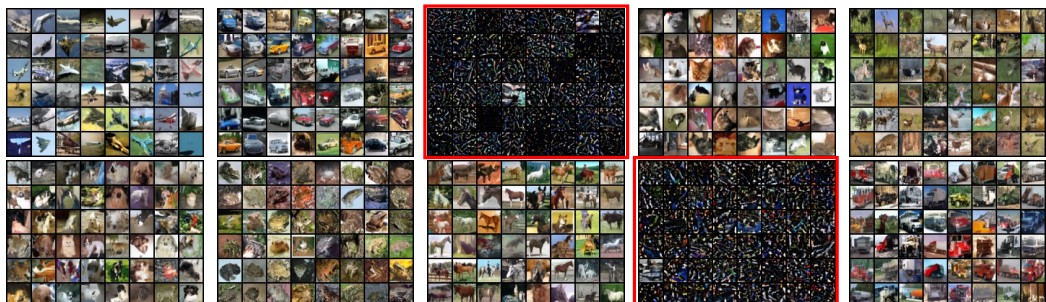

Figure 8: Images generated by our scrubbed conditional DDIM on CIFAR10 when we choose normal distribution $\hat{\epsilon}_t \sim \mathcal{N}(\mathbf{0.5}, \mathbf{I}_d)$. Images in the red solid box are generated by conditioning on the forgetting classes $\mathcal{C}_f$, others are generated by conditioning on the remaining classes $\mathcal{C}_r$.

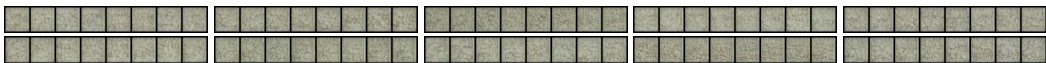

Figure 9: Images generated by our scrubbed conditional DDIM on CIFAR10. Sampling with noise from the uniform distribution $\hat{\mathbf{x}}_T \sim \mathcal{U}(\mathbf{0}, \mathbf{1})$.

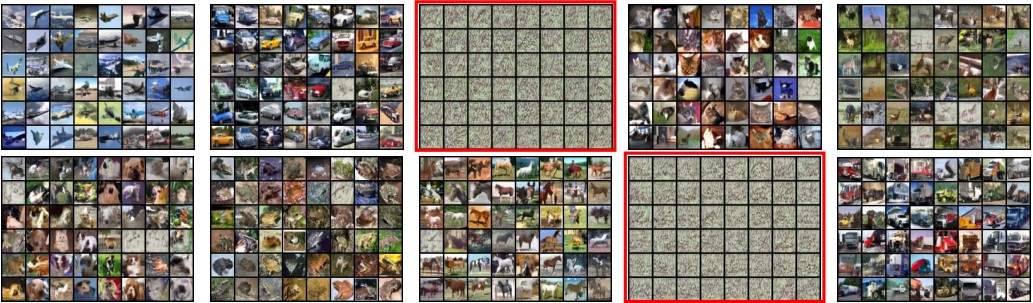

Figure 10: Images generated by our scrubbed conditional DDIM on CIFAR10 when $p_{uncond} \approx 0$.

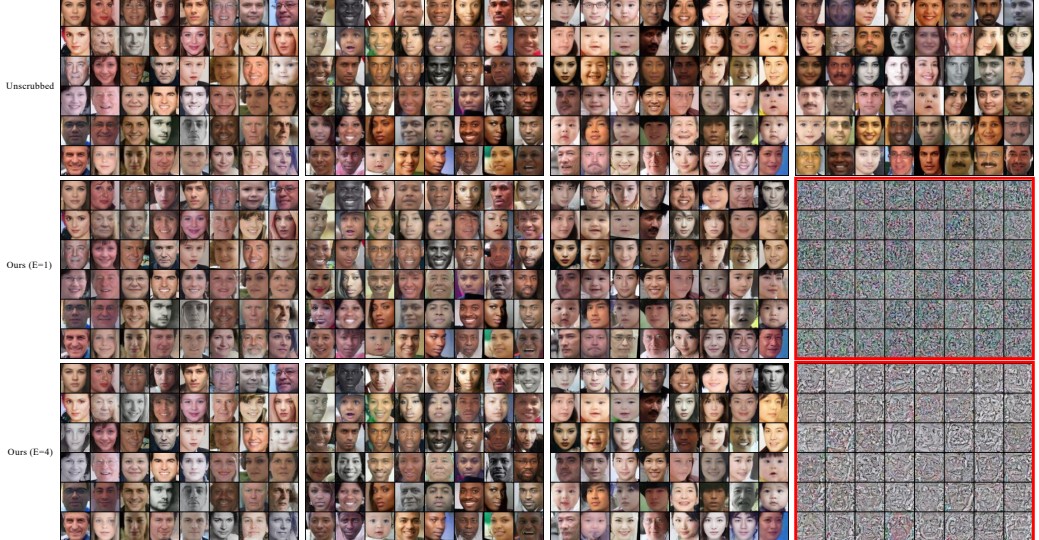

Figure 11: Images generated by conditional DDIM on UTKFace with different hyper-parameter $E$. Images in the red solid box are generated by conditioning on $\mathcal{C}_f$, others are generated by conditioning on $\mathcal{C}_r$. The larger the number $E$, the better the quality of generated images over $\mathcal{C}_r$.

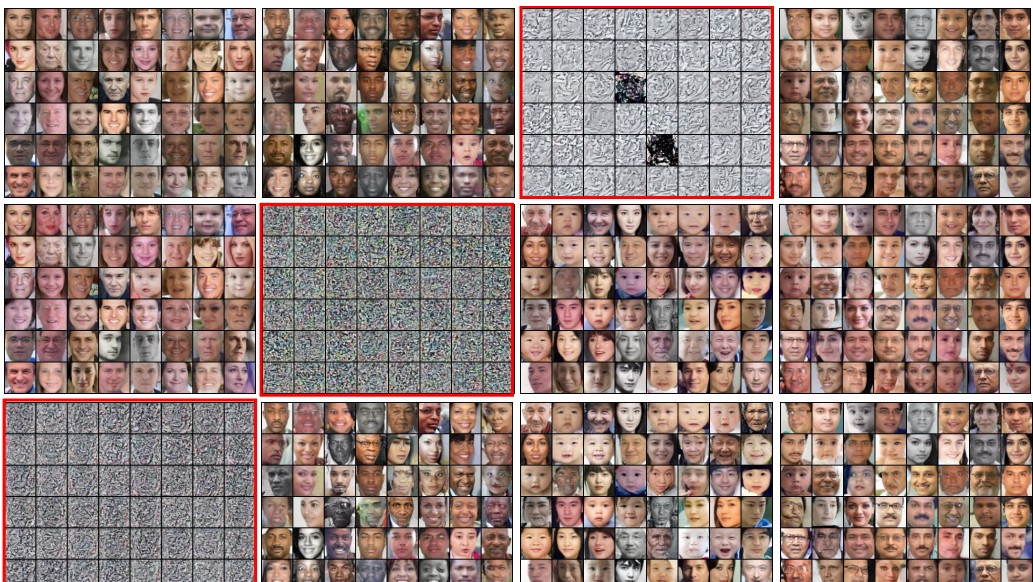

Figure 12: Images generated by our scrubbed conditional DDIM when unlearning different races (Top to Bottom: unlearning Asian, Black, and White, respectively). Images in the red solid box are generated by conditioning on $\mathcal{C}_f$, others are generated by conditioning on $\mathcal{C}_r$.

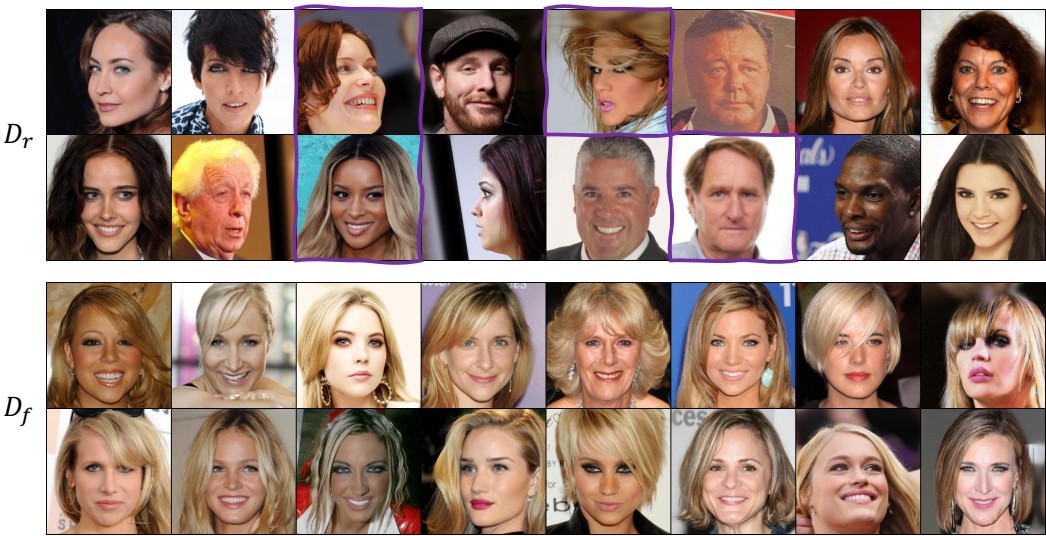

Figure 13: Examples from the remaining data $\mathcal{D}_r$ and forgetting data (Blond hair attribute) $\mathcal{D}_f$ on CelebA-HQ. Note that some examples in $\mathcal{D}_r$ (e.g., images in the purple solid box) have hair with a color that looks similar to the Blond hair attribute.

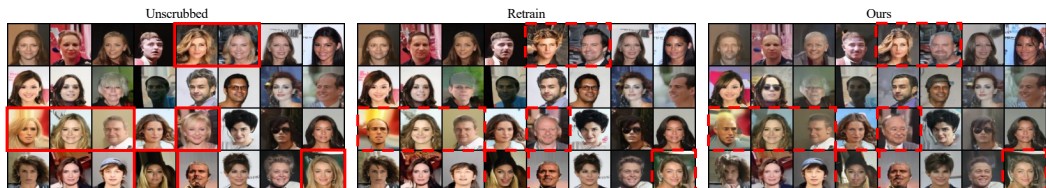

Figure 14: Images generated by unconditional DDIM on CelebA. We aim to unlearn the attribute of blond hair. Our unlearning algorithm obtains the results quite similar to those from the retrained model which is trained with the remaining data. Note that some images from the remaining data have hair attribute that looks like blond hair attribute as shown in Fig. 13.

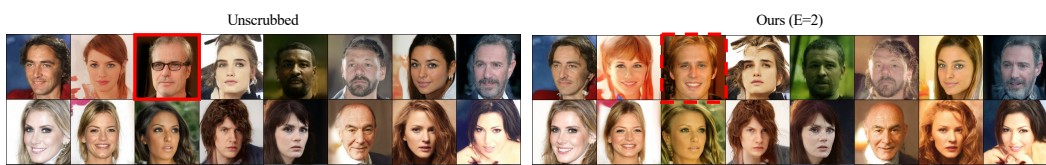

Figure 15: Images generated by the well-trained unconditional DDPM models from Hugging Face and our corresponding scrubbed models on CelebA-HQ. We aim to unlearn the Eyeglasses attribute.

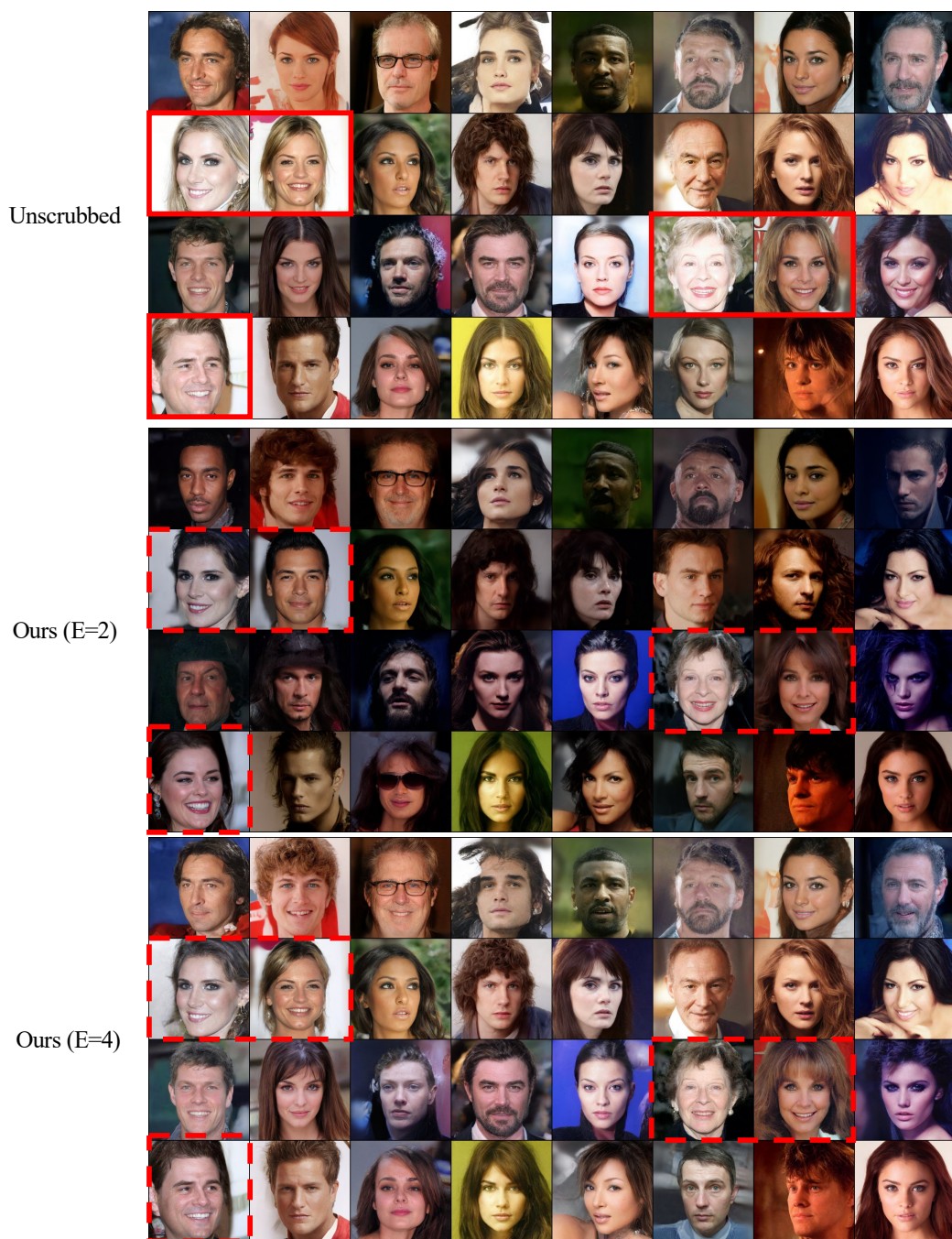

Figure 16: Images generated by the well-trained unconditional DDPM models from Hugging Face and our corresponding scrubbed models on CelebA-HQ. We aim to unlearn the Blond hair attribute.

Table 4: Results on CIFAR-10 with conditional DDIM, compared with simultaneously optimizing $\mathcal{L}(\boldsymbol{\theta}; \mathcal{D}_r) - \alpha\mathcal{L}(\boldsymbol{\theta}; \mathcal{D}_f)$ (denoted as SO). Generated examples are shown in Fig. 17. SO cannot achieve a good trade-off between erasing the influence of $\mathcal{D}_f$ and preserving model utility over $\mathcal{D}_r$.

| Method | FID over forgetting classes | | FID over remaining classes | | | | | | | |
|---|---|---|---|---|---|---|---|---|---|---|
| | $c=2\uparrow$ | $c=8\uparrow$ | $c=0\downarrow$ | $c=1\downarrow$ | $c=3\downarrow$ | $c=4\downarrow$ | $c=5\downarrow$ | $c=6\downarrow$ | $c=7\downarrow$ | $c=9\downarrow$ |
| Unscrubbed | 19.62 | 12.05 | 17.04 | 9.67 | 19.88 | 14.78 | 20.56 | 17.16 | 11.53 | 11.44 |
| Retrain | 152.39 | 139.62 | 17.39 | 9.57 | 20.05 | 14.65 | 20.19 | 17.85 | 11.63 | 10.85 |
| SO ($\alpha$=0.1) | 20.85 | 11.72 | 18.74 | 12.14 | 22.53 | 16.44 | 24.17 | 17.56 | 13.59 | 15.55 |
| SO ($\alpha$=0.3) | 33.33 | 22.87 | 20.22 | 12.05 | 24.12 | 21.00 | 26.18 | 21.57 | 14.24 | 15.00 |
| SO ($\alpha$=0.5) | 175.17 | 77.46 | 90.30 | 25.43 | 64.28 | 57.89 | 55.07 | 51.68 | 40.77 | 37.94 |
| *EraseDiff* (Ours) | 256.27 | 294.08 | 29.61 | 22.10 | 28.65 | 27.68 | 35.59 | 23.93 | 21.24 | 24.85 |

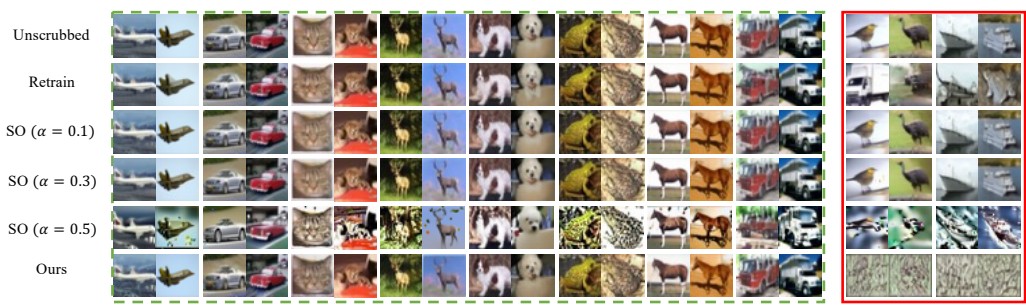

Figure 17: Images generated by conditional DDIM from Tab. 4. Images in the green dashed box are generated by conditioning on the remaining labels $\mathcal{C}_r$ and those in the red solid box are generated by conditioning on the forgetting classes $\mathcal{C}_f$.

Table 5: Results on CIFAR-10 trained with conditional DDIM, compared with separate optimization (Two-steps, denoted as TS). TS will perform $E_1$ epochs for the first step (ie., NegGrad), then perform $E_3$ epochs for the second step (ie., relearn using $\mathcal{D}_r$). Generated examples are shown in Fig. 18. TS cannot completely erase the influence of $\mathcal{D}_f$ on the model.

| Method | FID over forgetting classes | | FID over remaining classes | | | | | | | |
|---|---|---|---|---|---|---|---|---|---|---|
| | $c=2\uparrow$ | $c=8\uparrow$ | $c=0\downarrow$ | $c=1\downarrow$ | $c=3\downarrow$ | $c=4\downarrow$ | $c=5\downarrow$ | $c=6\downarrow$ | $c=7\downarrow$ | $c=9\downarrow$ |
| Unscrubbed | 19.62 | 12.05 | 17.04 | 9.67 | 19.88 | 14.78 | 20.56 | 17.16 | 11.53 | 11.44 |
| Retrain | 152.39 | 139.62 | 17.39 | 9.57 | 20.05 | 14.65 | 20.19 | 17.85 | 11.63 | 10.85 |
| TS (step 1, $E_1=10$) | 292.35 | 297.94 | 276.75 | 296.48 | 313.51 | 317.70 | 310.61 | 326.49 | 311.01 | 296.05 |
| TS (step 2, $E_2=50$) | 73.29 | 100.72 | 73.78 | 78.23 | 67.21 | 70.79 | 72.85 | 56.41 | 74.13 | 82.86 |
| TS (step 2, $E_2=100$) | 30.88 | 26.56 | 21.64 | 13.96 | 24.19 | 19.14 | 26.32 | 19.44 | 15.49 | 17.38 |
| *EraseDiff* (Ours) | 256.27 | 294.08 | 29.61 | 22.10 | 28.65 | 27.68 | 35.59 | 23.93 | 21.24 | 24.85 |

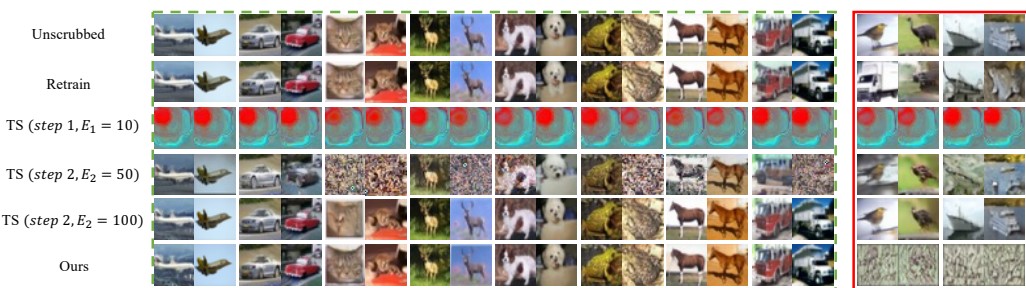

Figure 18: Images generated by conditional DDIM from Tab. 5. Images in the green dashed box are generated by conditioning on the remaining labels $\mathcal{C}_r$ and those in the red solid box are generated by conditioning on the forgetting classes $\mathcal{C}_f$.

Table 6: Results on CIFAR-10 trained with conditional DDIM. $\mathcal{D}_r'$: *EraseDiff* apply generated images to be the remaining data for the unlearning process.

| Method | FID over forgetting classes | | FID over remaining classes | | | | | | | |
|---|---|---|---|---|---|---|---|---|---|---|
| | $c=2\uparrow$ | $c=8\uparrow$ | $c=0\downarrow$ | $c=1\downarrow$ | $c=3\downarrow$ | $c=4\downarrow$ | $c=5\downarrow$ | $c=6\downarrow$ | $c=7\downarrow$ | $c=9\downarrow$ |
| Unscrubbed | 19.62 | 12.05 | 17.04 | 9.67 | 19.88 | 14.78 | 20.56 | 17.16 | 11.53 | 11.44 |
| Retrain | 152.39 | 139.62 | 17.39 | 9.57 | 20.05 | 14.65 | 20.19 | 17.85 | 11.63 | 10.85 |
| Finetune | 31.64 | 21.22 | 20.49 | 12.38 | 23.47 | 17.80 | 25.51 | 18.23 | 14.43 | 16.09 |
| NegGrad | 322.67 | 229.08 | 285.25 | 290.57 | 338.49 | 290.23 | 312.44 | 339.43 | 320.63 | 278.03 |
| BlindSpot | 349.60 | 335.69 | 228.92 | 181.88 | 288.88 | 252.42 | 242.16 | 278.62 | 192.67 | 195.27 |
| *EraseDiff* ($\mathcal{D}_r'$) | 298.60 | 311.59 | 33.01 | 24.09 | 34.23 | 34.79 | 45.51 | 38.05 | 24.59 | 28.10 |
| *EraseDiff* (Ours) | 256.27 | 294.08 | 29.61 | 22.10 | 28.65 | 27.68 | 35.59 | 23.93 | 21.24 | 24.85 |

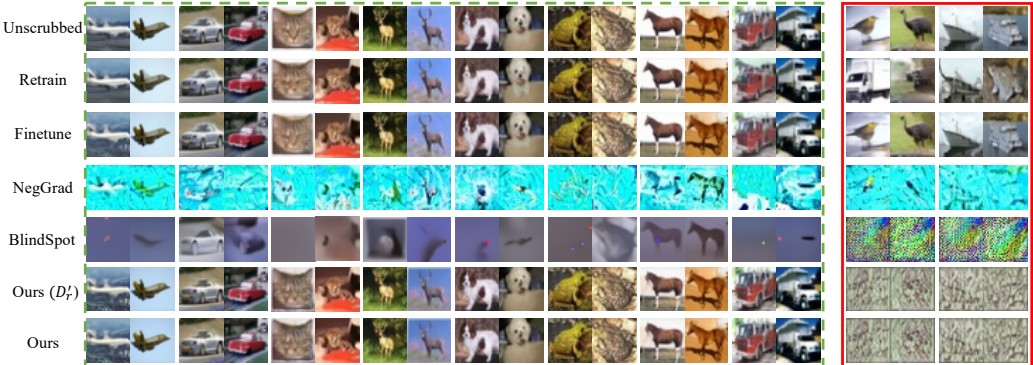

Figure 19: Images generated by conditional DDIM from Tab. 6. Images in the green dashed box are generated by conditioning on the remaining labels $\mathcal{C}_r$ and those in the red solid box are generated by conditioning on the forgetting classes $\mathcal{C}_f$.

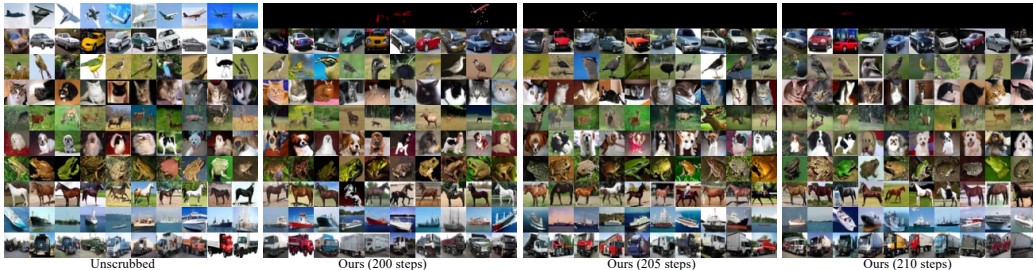

Figure 20: Conditional DDPM on CIFAR-10 when forgetting samples belonging to label '0'. Following Heng & Soh (2023) and using the well-trained model from Heng & Soh (2023), our method achieves a FID score of 8.93 at 210 steps, 8.83 at 205 steps, and 8.90 at 200 steps.

Table 7: Results of *EraseDiff* on CIFAR-10 with conditional DDIM. For each class, the FID score is computed over 5K generated images. Each row's forgetting classes are highlighted in orange.

| $\mathcal{C}_f$ | $c=0$ | $c=1$ | $c=2$ | $c=3$ | $c=4$ | $c=5$ | $c=6$ | $c=7$ | $c=8$ | $c=9$ |
|---|---|---|---|---|---|---|---|---|---|---|
| $c=2$ | 26.60 | 17.04 | 295.48 | 27.07 | 32.32 | 30.45 | 28.58 | 19.77 | 17.60 | 20.67 |
| $c=2,8$ | 29.61 | 22.10 | 256.27 | 28.65 | 27.68 | 35.59 | 23.93 | 21.24 | 294.08 | 24.85 |
| $c=5,6$ | 30.03 | 16.51 | 29.37 | 33.50 | 22.12 | 321.09 | 302.01 | 20.06 | 21.94 | 21.10 |
| $c=2,5,8$ | 24.02 | 15.59 | 288.01 | 26.06 | 19.31 | 296.79 | 21.25 | 15.87 | 206.61 | 21.56 |

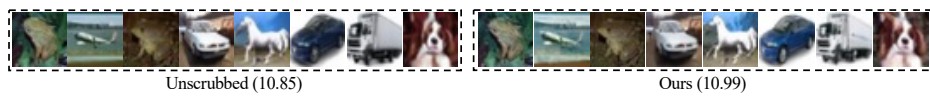

Figure 21: Unconditional DDPM on CIFAR-10 when forgetting randomly selected samples. 50K generated images by our scrubbed model have an FID score of 10.99, and the unscrubbed model has an FID score of 10.85.

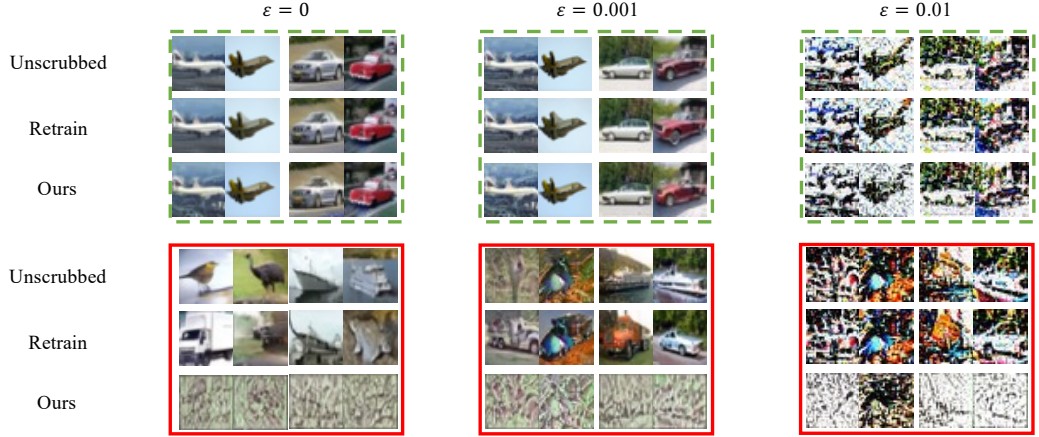

Figure 22: Generated examples when objected to FGSM Goodfellow et al. (2014) attack. Images in the green dashed box are generated by conditioning on the remaining labels $\mathcal{C}_r$ and those in the red solid box are generated by conditioning on the forgetting classes $\mathcal{C}_f$. With the step size $\epsilon$ increases, the quality of the generated images would decrease for all models. Note that our scrubbed model still doesn't contain information about the forgetting classes $\mathcal{C}_f$ in this setting.

Table 8: Results on UTKFace with conditional DDIM. SO: simultaneously optimizing $\mathcal{L}(\boldsymbol{\theta}; \mathcal{D}_r) - \alpha\mathcal{L}(\boldsymbol{\theta}; \mathcal{D}_f)$. Generated examples are shown in Fig. 23. *EraseDiff* achieves a better trade-off between erasing the influence of $\mathcal{D}_f$ and preserving model utility over $\mathcal{D}_r$ than SO.

| Method | FID over forgetting classes | FID over remaining classes | | |
|---|---|---|---|---|
| | $c = 3 \uparrow$ | $c = 0 \downarrow$ | $c = 1 \downarrow$ | $c = 2 \downarrow$ |
| Unscrubbed | 8.87 | 7.37 | 11.28 | 9.72 |
| SO ($\alpha$=0.05) | 216.35 | 14.09 | 15.73 | 15.62 |
| SO ($\alpha$=0.10) | 417.90 | 22.00 | 24.34 | 22.60 |
| *EraseDiff* (Ours) | 330.33 | 8.08 | 13.52 | 12.37 |

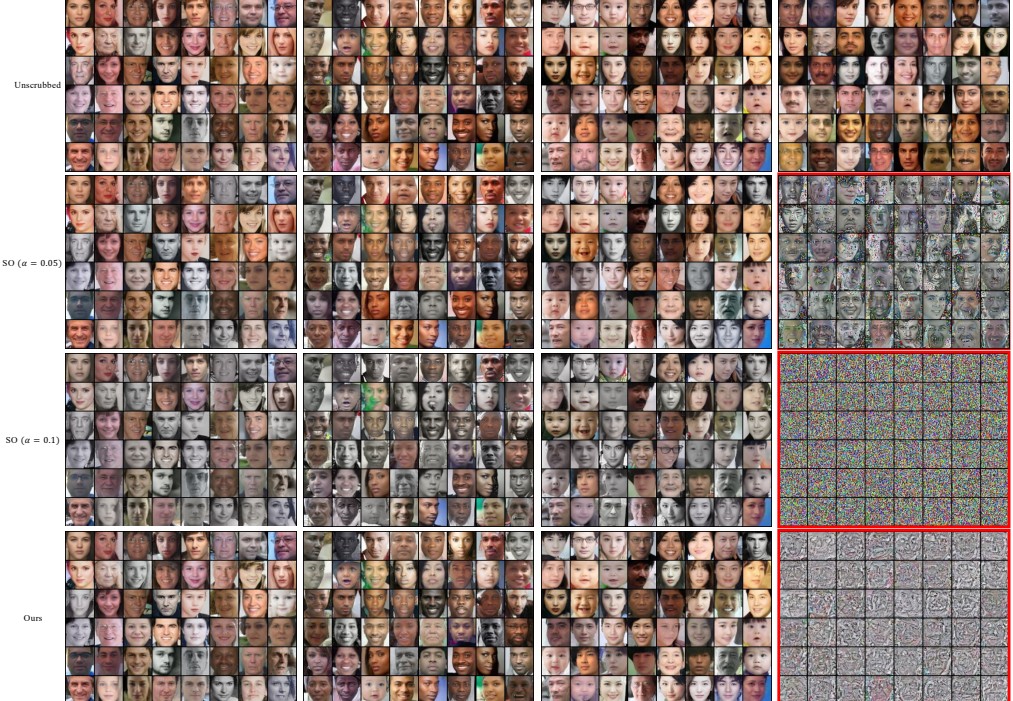

Figure 23: Images generated with conditional DDIM when unlearning the Indian celebrities from Tab. 8 (Top to Bottom: generated examples of the unscrubbed model, those of the model scrubbed by SO ($\alpha = 0.05$), those of the model scrubbed by SO ($\alpha = 0.10$), and those by our scrubbed model, respectively). Images in the red solid box are generated by conditioning on $\mathcal{C}_f$, others are generated by conditioning on $\mathcal{C}_r$. SO ($\alpha = 0.05$) cannot completely erase information about $\mathcal{C}_f$ and SO ($\alpha = 0.10$) has a significant drop in the quality of generated images.

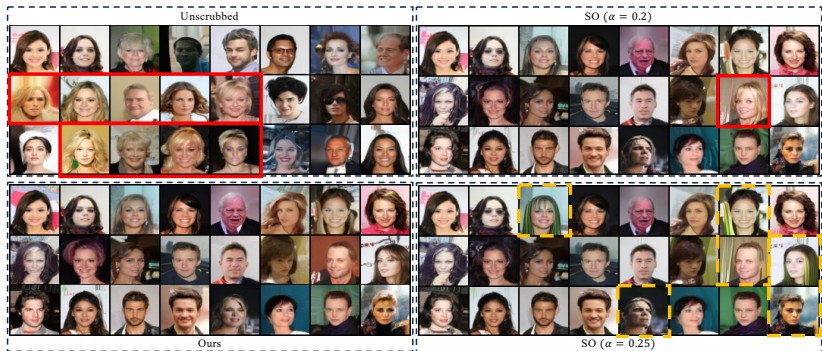

Figure 24: Images generated by unconditional DDIM on CelebA, focusing on the removal of the blond hair attribute. Images in the red solid box present the attribute of blond hair and those in the yellow dashed box display distortions. The FID score of the unscrubbed model, that of ours, that of SO ($\alpha = 0.2$), and that of SO ($\alpha = 0.25$) are 8.95, 10.70, 12.35, and 17.21 respectively.

## B DETAILED FORMULATION

Our formulation is

$$\boldsymbol{\theta}^* := \arg\min_{\boldsymbol{\theta}} \mathcal{F}(\boldsymbol{\theta}), \quad \text{where } \mathcal{F}(\boldsymbol{\theta}) = \mathcal{L}(\text{Alg}(\boldsymbol{\theta}, \mathcal{D}_f), \mathcal{D}_r) := \mathcal{F}(\boldsymbol{\theta}, \mathcal{D}_r) + \lambda \hat{f}(\boldsymbol{\theta}, \mathcal{D}_f), \quad (10)$$

We consider

$$h(\boldsymbol{\theta}, \phi) := \mathcal{F}(\boldsymbol{\theta}, \mathcal{D}_r) + \lambda \hat{f}(\boldsymbol{\theta}, \phi), \quad (11)$$

where $\hat{f}(\boldsymbol{\theta}, \phi) = f(\phi, \mathcal{D}_f) - f(\boldsymbol{\theta}, \mathcal{D}_f)$, then

$$\text{Alg}(\boldsymbol{\theta}, D_f) = \phi^*(\boldsymbol{\theta}) = \text{argmin}_{\phi} h(\boldsymbol{\theta}, \phi) = \arg\min_{\phi} \hat{f}(\boldsymbol{\theta}, \phi) = \arg\min_{\phi|\boldsymbol{\theta}} f(\phi, \mathcal{D}_f), \quad (12)$$

where $\phi \mid \boldsymbol{\theta}$ means $\phi$ is started from $\boldsymbol{\theta}$ for its updates. Finally, we reach

$$\min_{\boldsymbol{\theta}} \min_{\phi \in \phi^*(\boldsymbol{\theta})} h(\boldsymbol{\theta}, \phi) = \min_{\boldsymbol{\theta}} \min_{\phi \in \text{Alg}(\boldsymbol{\theta}, D_f)} h(\boldsymbol{\theta}, \phi). \quad (13)$$

We can characterize the solution of our algorithm as follows:

**Theorem 1** (Pareto optimality)**.** *The stationary point obtained by our algorithm is Pareto optimal.*

*Proof.* Let $\theta^*$ be the solution to our problem. Because given the current $\theta_s$, in the inner loop, we find $\phi_s^K$ to minimize $\hat{f}(\phi, \mathcal{D}_f) = f(\theta_s, \mathcal{D}_f) - f(\phi, \mathcal{D}_f)$. Assume that we can update in sufficient number of steps $K$ so that $\phi_s^K = \phi^*(\theta_s) = \arg\min_{\phi|\theta_s} \hat{f}(\phi, \mathcal{D}_f) = \arg\min_{\phi|\theta_s} f(\phi, \mathcal{D}_f)$. Here $\phi \mid \theta_s$ means $\phi$ is started from $\theta_s$ for its updates.

The outer loop aims to minimize $\mathcal{F}(\theta, \mathcal{D}_f) + \lambda \hat{f}(\phi^*(\theta), \mathcal{D}_r)$ whose optimal solution is $\theta^*$. Note that $\hat{f}(\phi^*(\theta), \mathcal{D}_r) \geq 0$ and it decreases to 0 for minimizing the above sum. Therefore, $\hat{f}(\phi^*(\theta^*), \mathcal{D}_r) = 0$. This further means that $\hat{f}(\theta^*, \mathcal{D}_f) = \hat{f}(\phi(\theta^*), \mathcal{D}_f)$, meaning that $\theta^*$ is the current optimal solution of $\hat{f}(\phi, \mathcal{D}_f)$ because we cannot update further the optimal solution. Moreover, we have $\theta^*$ as the local minima of $\mathcal{F}(\theta, \mathcal{D}_f)$ because $\hat{f}(\phi^*(\theta^*), \mathcal{D}_f) = 0$ and we consider a sufficiently small vicinity around $\theta^*$. $\qquad\square$

