# OpenReview forum: "EraseDiff: Erasing Data Influence in Diffusion Models"
_ICLR.cc/2024/Conference — Submitted to ICLR 2024_

### Official Review · Reviewer_oTvb · 2023-10-28

**Soundness:** 3 good
**Presentation:** 3 good
**Contribution:** 2 fair
**Rating:** 5
**Confidence:** 4

**Summary:**

This paper introduces an unlearning algorithm for diffusion models in order to mitigate the concerns about data memorization. With the goal of scrubbing the information for certain classes, this paper proposes to maximize the variational bound on the sub-dataset under those classes while also minimizing the objective function on the remaining data.  To solve the constrained optimization, the author proposed to view it as a bi-level optimization and perform maximization and minimization alternatively to erase the information while maintaining the model quality. The proposed method is simple and effective, as demonstrated by numerical experiments. My major concern is that the practical performance fails to outperform in both forgetting and remaining classes. The author may claim their method achieves a better tradeoff, which is not fully supported by the experiments and could be subjective. It would be better to demonstrate some kind of “optimality” either theoretically or empirically to back up this simple idea.

**Strengths:**

The idea is simple and effective. The effectiveness is shown via numerical experiments, where the proposed method achieves an arguably better tradeoff between forgetting some certain classes and remaining to perform well in other classes.

**Weaknesses:**

Is there a theoretical guarantee to do the optimization alternatively? It seems not necessary in this setting, as we may do it completely in a separate way, i.e., do the maximization first to completely scrub the forgetting classes and then fine-tune to make up the performance on the remaining dataset. Which way would be better, and why?

We also see a drop in the quality of generated figures from Table 2, and the proposed model is worse than finetune from Table 3.

**Questions:**

Minor typo: page 3, “the variational bond” above equation (1).

---

> ### Author Response · Authors · 2023-11-20
> **Response to Reviewer oTvb**
>
> Thank you for your thoughtful feedback and suggestions, which we have now incorporated into the new revision of the manuscript. **New results are included from pages 20 to 22, with changes highlighted for ease of tracking.** We address your questions below. We are happy to continue the discussion in case of follow-ups.
>
> **Q5.1 Is there a theoretical guarantee to do the optimization alternatively? It seems not necessary in this setting, as we may do it completely in a separate way, i.e., do the maximization first to completely scrub the forgetting classes and then fine-tune to make up the performance on the remaining dataset. Which way would be better, and why?**
>
> The nature of first forgetting and then retraining suggests possible degradation as either the model is seriously damaged by the forgetting step or even the possibility of relearning the forgetting data as a result of retraining (especially for the classes with large resemblances). To put this to test, we performed an experiment on CIFAR-10 by first performing NegGrad and then relearning using the remaining data. Results, denoted as Two-steps, are shown in Table 5 and Figure 18 on page 20 in the revised manuscript. We observe significant deterioration in generating $\mathcal{C}_r$, while improvement in scrubbing $\mathcal{C}_f$. This formulation is observed to be even worse than the multi-task formulation. Interestingly, we observe that the generated images for $\mathcal{C}_f$ may contain relevant information after retraining.
> Overall, our proposed formulation leads to a robust and simple optimization framework.
> Thanks for the suggestion, further studies on effective disjoint optimization remain as future work which we have reflected in the conclusion of our revised manuscript.
>
> **Q5.2 We also see a drop in the quality of generated figures from Table 2, and the proposed model is worse than finetune from Table 3.**
>
> Finetune has a better FID on remaining classes $\mathcal{C}_r$ but fails to scrub the information about the forgetting classes $\mathcal{C}_f$, as shown in Figure 4 on page 9. Compared with other methods, our method achieves the best trade-off between removing information about $\mathcal{C}_f$ and preserving model utility over $\mathcal{C}_r$ with good efficiency.
>
> **Q5.3 Minor typo: page 3, “the variational bond” above equation (1).**
>
> Thanks! We have since corrected it.

---

### Official Review · Reviewer_LbvE · 2023-10-30

**Soundness:** 3 good
**Presentation:** 4 excellent
**Contribution:** 4 excellent
**Rating:** 8
**Confidence:** 3

**Summary:**

The paper propose the first(up to their knowledge) unlearning algorithm for diffusion models in order to let the models learn to erase the learning effects by some specific training data and still remember the distribution of other training data, without retraining the models from scratch. Such algorithm help protect privacy, prevent misuse, mitigate or erase bad impact by some undesirable training data but preserve model utility with respect to remaining training data.

**Strengths:**

1. This paper proposes a new strategy to finetune diffusion model to eliminate effects of some training data.
2. This includes experiments comparing the effectiveness of their model on both label-conditional and unconditional diffusion models, showing that by the algorithm provided, a pretrained model can actually forget designated data and preserve the rest.
3. Problem addressing, motivation, method(learning objective design and finetune pipeline), experiments are clear.
4. This paper demonstrate efficacy and efficiency to scrub diffusion model and such contribution can be significant if more and more privacy concerns are addressed on those generative models.

**Weaknesses:**

The evaluation is based too much on image label, which might only be a small subset of the potential problem cases.
For class conditional diffusion model, it makes sense to run the finetune algorithm to forget all training images for specific class, it is like a reverse-process of few-shot learning.
For unconditional diffusion model, the training data still provide image class(but diffusion model is not able to access label during training or testing). Then, by erasing some all data from a specific class, the diffusion model does not actually "forget well" about those data. Such case is severe especially for alike classes, and for human face dataset like CelebA.
Most importantly, if the elements in forgetting subset do NOT share enough common property(for example, a photographer want you to erase all photos she took, but those photos are of huge variety with many semantics, despite "her special style" can be labeled, this is often not a label available in original model training), in that case, it is hard to evaluate how good the model forgets such subset.

**Questions:**

Suggestions:
1. Design more experiment(and have more discussion) on random(or not so well-labeled) subset, demonstrate model efficacy.
2. Evaluating unconditional diffusion model utility with respect to specific sub-dataset can be hardly well-defined, so try to narrow-down and specify the problem you want to tackle in a more guided way(such as to only some specific conditional diffusion model).
3. Try more types of condition(or context).

---

> ### Author Response · Authors · 2023-11-20
> **Response to Reviewer LbvE**
>
> Thank you for your thoughtful feedback and suggestions, which we have now incorporated into the new revision of the manuscript. **New results are included from pages 20 to 22, with changes highlighted for ease of tracking**. We address your questions below. We are happy to continue the discussion in case of follow-ups.
>
> **Q4.1 Design more experiment (and have more discussion) on random (or not so well-labeled) subset, demonstrate model efficacy.**
>
> Thanks for the suggestion. We conducted an additional experiment to perform sample unlearning on an unconditional DDPM with pre-trained weights from HuggingFace. Examples of generated images from the unscrubbed model and our scrubbed model are shown in Figure 21 on page 22. FID scores over 50K images are 10.85 and 10.99 for the unscrubbed model and our scrubbed model, respectively.
> We can see that the quality of generated images only drops by around 0.14 in terms of the FID score. It is true that it is hard to verify the unlearning effect in this setting. We'll investigate how to evaluate in a sample-wise unlearning setting in future work.
>
> **Q4.2 Evaluating unconditional diffusion model utility with respect to specific sub-dataset can be hardly well-defined, so try to narrow-down and specify the problem you want to tackle in a more guided way (such as to only some specific conditional diffusion model).**
>
> Thanks for the suggestion. We clarified this in the conclusion of our revised manuscript.
>
> **Q4.3 Try more types of condition( or context).**
>
> Thanks for the suggestion. More generated image examples with different settings can be found in the Appendix (unlearning different classes for CIFAR-10 and UTKFace, and different attributes on CelebA and CelebA-HQ). We have since added Table 7 on page 21 to show the performance in different settings.

---

### Official Review · Reviewer_JxPy · 2023-10-31

**Soundness:** 1 poor
**Presentation:** 2 fair
**Contribution:** 1 poor
**Rating:** 3
**Confidence:** 4

**Summary:**

This paper focuses on the development of an algorithm called EraseDiff, which aims to address the privacy risks and data protection regulations associated with diffusion models. These models, known for their high-quality output and ease of use, pose concerns related to privacy, memorization of training data, generation of inappropriate content, and potential violation of data ownership and copyright laws. The proposed algorithm formulates unlearning as a bi-level optimization problem, with the goal of scrubbing the information associated with forgetting data from diffusion models without the need for retraining the entire system. The algorithm is evaluated in various scenarios, including the removal of classes, attributes, and races from different datasets, and it demonstrates improved performance compared to baseline methods.

**Strengths:**

1. This paper focuses on a crucial question: the issue of privacy within the diffusion model.

**Weaknesses:**

1. The formulation of the diffusion model unlearning problem in this work seems unconventional. Both the inner and outer objectives aim to optimize the model parameters. As such, it can be naturally defined as a multitask problem. One task seeks to preserve the utility of the diffusion model for the remaining data, while the other aims to eliminate the information related to the data slated for removal.

2. The inner objective of this work strives to make the diffusion model incapable of generating meaningful images corresponding to C_f. The rationale behind defining sample unlearning in this manner is unclear. A more intuitive approach would be to ensure that the model, when trained with both the remaining and the forgetting data, has parameters equivalent to those obtained when trained solely on the remaining data after the unlearning process.

**Questions:**

N/A

---

> ### Author Response · Authors · 2023-11-20
> **Response to Reviewer JxPy**
>
> Thank you for your thoughtful feedback and suggestions, which we have now incorporated into the new revision of the manuscript. **New results are included from pages 20 to 22, with changes highlighted for ease of tracking**. We address your questions below. We are happy to continue the discussion in case of follow-ups.
>
> **Q3.1 The diffusion unlearning can be naturally defined as a multitask problem.**
>
> Indeed, this formulation is possible; however, it would not perform better than our proposal empirically.
> To provide further insights, we performed extra experiments on the CIFAR-10 dataset using conditional DDIM (please see Tables 4 and 5, Figures 17 and 18 on page 20 in the revised manuscript). Formulation as a multi-task problem and solving it via scalarization as a single-level optimization is shown as SO in Table 4 and Figure 17. We observed that this would lead to a significant drop in the image quality on $\mathcal{C}_r$ when boosting to forget $\mathcal{C}_f$ by large.
> As shown in Figure 17 ($\alpha=0.5$), when the generated images yield coarse information for $\mathcal{C}_f$, the quality of generated images for $\mathcal{C}_r$ drops significantly.
> Overall, we believe formulating as vanilla multi-task problem is not beneficial. Thanks for the suggestion; further studies on effective multi-task formulation remain as future work, which we have reflected in the conclusion of our revised manuscript.
>
> **Q3.2 $\textbf{I.}$ The inner objective of this work strives to make the diffusion model incapable of generating meaningful images corresponding to $\mathcal{C}_f$. The rationale behind defining sample unlearning is unclear. $\textbf{II.}$ A more intuitive approach would be to ensure that the model, when trained with both the remaining and the forgetting data, has parameters equivalent to those obtained when trained solely on the remaining data after the unlearning process.**
>
> Our objective is defined with respect to the forgetting data $\mathcal{D}_f$ and the remaining data $\mathcal{D}_r$, $\mathcal{D}_f$ could be randomly selected from the training data, i.e., sample unlearning.
> We didn't evaluate sample-wise unlearning because, currently, it is hard to evaluate the unlearning effect when performing sample-wise unlearning for diffusion models, as Reviewer ${\color{orange}LbvE}$ mentioned.
> Per your comment, we conducted an experiment to perform sample unlearning on an unconditional DDPM with pre-trained weights from HuggingFace (i.e., the forgetting data $\mathcal{D}_f$ is randomly selected from). Examples of generated images from the unscrubbed model and our scrubbed model are shown in Figure 21 on page 22 in the revised manuscript. FID scores over 50K images are 10.85 and 10.99 for the unscrubbed model and our scrubbed model respectively. We can see that the quality of generated images only drops by around 0.14 in terms of the FID score.
>
> It's true that the scrubbed model with unlearning algorithms should be equivalent to the retrained model which uses the remaining data to retrain from scratch, which is the ideal unlearning algorithm. However, the removal of pertinent data followed by retraining diffusion models from scratch demands substantial resources and is often deemed impractical. Therefore, the model retrained solely on the remaining data is unknown to the unlearning algorithms. Instead, approximate unlearning algorithms (e.g., [1-2] and ours) achieve model scrubbing without the need for retraining. This is accomplished by updating the model parameters in a negative direction relative to the samples or concepts intended for erasure.
>
> We evaluated the Weight Distance (WD) between the scrubbed models and the retrained model (please see Table 3 on page 8) in experiments.
> Table 3 reveals that although the NegGrad method has a Weight Distance (WD) of 1.3533, it is unable to preserve model utility. In contrast, our approach attains the smallest WD among effective unlearning algorithms, i.e., 1.3534. Importantly, it successfully erases information about the forgetting classes $\mathcal{C}_f$, while concurrently preserving data regarding the remaining classes $\mathcal{C}_r$.
>
> ---
> >[1] Rohit Gandikota, Joanna Materzynska, Jaden Fiotto-Kaufman, and David Bau. Erasing concepts from diffusion models. arXiv preprint arXiv:2303.07345, 2023.
> >
> >[2] Heng, Alvin, and Harold Soh. "Selective Amnesia: A Continual Learning Approach to Forgetting in Deep Generative Models." arXiv preprint arXiv:2305.10120 (2023).

---

> > ### Comment · Reviewer_JxPy · 2023-11-22
> >
> > Dear Authors,
> >
> > Thank you all for your responses.
> >
> > 1. The paper attempts to treat the optimization of model parameters as both the inner and outer objectives in a bi-level optimization framework. However, according to the conventional definition of bi-level optimization, as referenced in [1], this approach is not a bi-level optimization problem.
> >
> >
> > 2. The authors of this work suggest that alternative formulations, like the multitask formulation I previously proposed, are not as effective based on empirical evidence. But empirical evidence alone might not be convincing enough. Is there any theoretical support for the bi-level approach being better than other methods, as Reviewer oTvb has asked? If theoretical analysis is lacking, a detailed comparison of the bi-level method against other approaches would be beneficial. This is particularly relevant given that several studies (referenced as [2] and [3],
> > [2] can achieve their goal with Plug-and-play Loss and Diffusion prior loss, [3] can achieve their goal with the proposed loss and prior preservation loss) have shown success with different strategies, the multitask formulation. The current empirical results, which mainly focus on CIFAR-10, appear narrow in scope when compared to these other studies. It's also worth noting that bi-level problems typically seem more complex than multitask ones.
> >
> >
> > At this stage, I am inclined to maintain my initial rating and reinforce my judgment with greater confidence.
> >
> >
> > Best regards,
> >
> > --
> >
> > [1] Chen, C., Chen, X., Ma, C., Liu, Z., & Liu, X. (2022). Gradient-based bi-level optimization for deep learning: A survey. arXiv preprint arXiv:2207.11719.
> >
> > [2] Song, J., Zhang, Q., Yin, H., Mardani, M., Liu, M. Y., Kautz, J., ... & Vahdat, A. (2023). Loss-Guided Diffusion Models for Plug-and-Play Controllable Generation.
> >
> > [3] Ruiz, N., Li, Y., Jampani, V., Pritch, Y., Rubinstein, M., & Aberman, K. (2023). Dreambooth: Fine tuning text-to-image diffusion models for subject-driven generation. In Proceedings of the IEEE/CVF Conference on Computer Vision and Pattern Recognition (pp. 22500-22510).

---

> ### Author Response · Authors · 2023-11-23
> **Response to Reviewer JxPy (1/3)**
>
> We thank Reviewers ${\color{cyan}JxPy}$ and provide further insights per the comments.
>
> **Q1. According to the conventional definition of bi-level optimization, as referenced in [1], this approach is not a bi-level optimization problem.**
>
> To be precise, a bi-level optimization is a hierarchical mathematical program where the feasible region of one optimization task is constrained by the solution of another task (see [6] for a detailed description). Our proposed formulation aligns with this definition. We aim to erase the influence of forgetting data $\mathcal{D}_f$ ($\min Alg(\theta, \mathcal{D}_f)$) while simultaneously preserving the model utility over the remaining data $\mathcal{D}_r$ ($\min \mathcal{F}(\theta)$). Also, we note that not all bi-level optimization problems necessitate distinct parameter sets (if ${\color{cyan}JxPy}$ refers to the lack of that in our formulation). A prominent example is Model-Agnostic Meta-Learning (MAML) [4]. MAML's objective is:
>
> $$
> \quad \quad \theta_{ML}^{*} := \text{argmin}\_{\theta}  \mathcal{F}(\theta), \hspace{1em} \text{where} \hspace{0.5em} \mathcal{F}(\theta) = \frac{1}{M}\sum_{i=1}^M \mathcal{L}(\text{Alg}(\theta, \mathcal{D}_i^{tr}), \mathcal{D}_i^{test}), \tag{1}
> $$
> with $M$ denoting the number of meta-training tasks. The objective aims to achieve low generalization error on $\mathcal{D}_i^{test}$, while the inner objective is to learn knowledge of $\mathcal{D}_i^{tr}$. The objective of our method reads as follows:
>
> $$
> \quad \quad  \theta^{*} := \text{argmin}\_{\theta} \mathcal{F}(\theta), \hspace{1em} \text{where} \hspace{0.5em} \mathcal{F}(\theta) = \mathcal{L}(\text{Alg}(\theta, \mathcal{D}_f), \mathcal{D}_r):= \mathcal{F}(\theta, \mathcal{D}_r) + \lambda \hat{f}(\theta, \mathcal{D}_f), \tag{2}
> $$
> where $\hat{f}(\theta, \mathcal{D}_f):=f(\theta, \mathcal{D}_f)-\text{min}\_{\phi|\theta}f(\phi, \mathcal{D}_f)$ represents the delta erasing gap. Specifically, $f(\theta, \mathcal{D}_f)$ indicates the erasing loss at $\theta$, while $\text{min}\_{\phi|\theta}f(\phi, \mathcal{D}_f)$ specifies the optimal delta erasing gap we can yield if we depart from $\phi = \theta$ and minimize the erasing loss to its minimum. Here, note that we define $\text{Alg}(\theta, \mathcal{D}_f):= \text{argmin}\_{\phi|\theta} f(\phi, \mathcal{D}_f)$.
>
> Moreover, the above model-agnostic optimization problem states that we aim to reach the optimal model $\theta^*$ such that it can minimize the retaining loss $\mathcal{F}(\theta, \mathcal{D}_r)$ for $\mathcal{D}_r$, while also minimizing the delta erasing gap $\hat{f}(\theta, \mathcal{D}_f)$, i.e., from $\theta^*$, even if we try our best to erase $\mathcal{D}_f$, we cannot erase much.
>
> Note that both the inner and outer objectives in our formulation and MAML are directed at optimizing the model parameters.
> Our formulation distinction from MAML lies in its focus: we are not seeking a model adaptable to unlearning but one that effectively erases the influence of data points on the model.
> We hope this establishes the missing link and we have revised our manuscript to further clarify and strengthen this aspect in the related work section.

---

> ### Author Response · Authors · 2023-11-23
> **Response to Reviewer JxPy (2/3)**
>
> **Q2. Empirical evidence alone might not be convincing enough. Is there any theoretical support for the bi-level approach being better than other methods?**
>
> We are not aware of theoretical studies in that regard. However, we can characterize the solution of our algorithm as follows:
>
> $\textbf{Theorem 1}$ (Pareto optimality). $\quad \textit{The stationary point obtained by our algorithm is Pareto optimal.}$
>
> $\textit{proof.}$
>
> Let $\theta^\ast$ be the solution to our problem. Because given the current $\theta_s$, in the inner loop, we find $\phi^K_s$ to minimize $\hat{f}(\phi, \mathcal{D}_f)=f(\theta_s, \mathcal{D}_f)-f(\phi, \mathcal{D}_f)$. Assume that we can update in sufficient number of steps $K$ so that $\phi^K_s= \phi^*(\theta_s) = \text{argmin}\_{\phi \mid \theta_s} \hat{f}(\phi, \mathcal{D}_f) = \text{argmin}\_{\phi\mid\theta_s} f(\phi, \mathcal{D}_f)$. Here $\phi \mid \theta_s$ means $\phi$ is started from $\theta_s$ for its updates.
>
> The outer loop aims to minimize $\mathcal{F}(\theta, \mathcal{D}_f) + \lambda \hat{f}(\phi^*(\theta), \mathcal{D}_r)$ whose optimal solution is $\theta^\ast$. Note that $\hat{f}(\phi^*(\theta), \mathcal{D}_r)\geq 0$ and it decreases to $0$ for minimizing the above sum. Therefore, $\hat{f}(\phi^*(\theta^*), \mathcal{D}_r)=0$. This further means that $\hat{f}(\theta^*, \mathcal{D}_f) = \hat{f}(\phi(\theta^*), \mathcal{D}_f)$, meaning that $\theta^*$ is the current optimal solution of $\hat{f}(\phi,\mathcal{D}_f)$ because we cannot update further the optimal solution. Moreover, we have $\theta^*$ as the local minima of $\mathcal{F}(\theta,\mathcal{D}\_{f})$ because $\hat{f}(\phi^*(\theta^*),\mathcal{D}\_{f})=0$ and we consider a sufficiently small vicinity around $\theta^*$.
> $\square$
>
> Furthermore and as discussed in [1] and [6], multi-task learning (MTL) could also be expressed as a bi-level optimization problem (\eg, MAML [4]). In our case, simultaneously optimizing the two objectives, as scalarization of an MTL, is indeed an instantiation of our proposed algorithm. Consider Eq.(9) in the main paper, repeated below for the comfort of the reviewer.
> $$
> \quad \quad \theta^* = \theta - \zeta( \nabla\_{\theta}\mathcal{F}(\theta, \mathcal{D}\_{rs}) + \lambda \nabla\_{\phi}\hat{f}(\phi, \mathcal{D}\_f) ).  \tag{3}
> $$
> Here, $\hat{f}\phi, \mathcal{D}\_f) = f(\phi, \mathcal{D}\_f) - f(\phi^K, \mathcal{D}\_f)$ and in the inner loop, $\phi = \theta$. This shows that if $K=0$, then our solution recovers
> $$
>  \quad \quad  \theta^* = \theta - \zeta( \nabla\_{\theta}\mathcal{F}(\theta, \mathcal{D}\_{rs}) + \lambda \nabla\_{\theta} f(\theta, \mathcal{D}_f) ).  \tag{4}
> $$
> This is indeed the scalarization of MTL. Checking Figure 2 (a) in [4], Figure 2 in [5], and Figure 6 in our paper demonstrate that increasing the number of inner update steps (i.e., $K$ in our case) could improve the resulting model performance.
>
> Finally, while we acknowledge the importance of optimization, whether it be bi-level or other forms, it serves more as a tool to realize our main objective, the formulation of unlearning in diffusion models (DMs). Therefore, the lack of theoretical analysis on the optimization form should not overshadow the main theory (i.e., formulation of unlearning in DMs) our work offers. In essence, our paper addresses a fundamental problem by introducing new theories related to formulating unlearning, coupled with an engineering solution that involves developing the form of optimization. To overlook these aspects due to a lack of theoretical analysis on whether bi-level optimization is the best choice would not accurately represent the full scope and innovation of our work.

---

> > ### Author Response · Authors · 2023-11-23
> > **Response to Reviewer JxPy (3/3)**
> >
> > **Q3. If theoretical analysis is lacking, a detailed comparison of the bi-level method against other approaches would be beneficial.**
> >
> > Per your comment, we have revised our work, and the current version includes additional results on the UTKFace and CelebA datasets.
> > The updated analysis, as detailed in Table 8, Figures 23 and 24 on page 23, shows that our algorithm achieves a better trade-off between erasing information about $\mathcal{C}_f$ and maintaining model utility over $\mathcal{C}_r$ compared to Single Level Optimization (SO), i.e., the scalarization form of MTL. Specifically, on UTKFace, when the FID score of SO ($\alpha=0.05$) $\mathcal{C}_r$ is, on average, about 4 points higher than ours (in all classes, our solution is better on $\mathcal{C}_r$). Also, we note that on $\mathcal{C}_f$, our solution is significantly better than SO (FID of 330.33 compared to 216.35 on the forgetting data). Increasing the combination weight to $\alpha=0.10$ improves the FID for the forgetting data but drastically deteriorates the quality on $\mathcal{C}_r$.
> > When performing concept unlearning on the CelebA dataset, the FID scores of the unscrubbed model, that of ours, that of SO ($\alpha=0.2$), and that of SO ($\alpha=0.25$) are 8.95, 10.70, 12.35, and 17.21, respectively. Both SO ($\alpha=0.25$) and ours successfully removed the blond hair attribute; however, the images generated by SO ($\alpha=0.25$) are severely distorted.
> >
> > **Q4. This is particularly relevant given that several studies (referenced as [2] and [3], [2] can achieve their goal with Plug-and-play Loss and Diffusion prior loss.**
> >
> > Our solution enables us to equip any plug-and-play loss function into the unlearning procedure easily. This flexibility is achieved by changing the outer loop to incorporate plug-and-play loss functions, enhancing the quality of generation as needed. As a matter of fact, your comment highlights another advantage of our formulation and we are grateful for that. By executing unlearning in the inner loop, we effectively isolate it from interfering with the generation process. This separation allows our framework to seamlessly incorporate other diffusion solutions. We have updated the related work section in the revised manuscript to reflect this adaptability and compatibility with existing diffusion solutions.
> >
> > ---
> > >[1] Chen, C., Chen, X., Ma, C., Liu, Z., Liu, X. (2022). Gradient-based bi-level optimization for deep learning: A survey. arXiv preprint arXiv:2207.11719.
> > >
> > >[2] Song, J., Zhang, Q., Yin, H., Mardani, M., Liu, M. Y., Kautz, J., ... Vahdat, A. (2023). Loss-Guided Diffusion Models for Plug-and-Play Controllable Generation.
> > >
> > >[3] Ruiz, N., Li, Y., Jampani, V., Pritch, Y., Rubinstein, M., Aberman, K. (2023). Dreambooth: Fine tuning text-to-image diffusion models for subject-driven generation. In Proceedings of the IEEE/CVF Conference on Computer Vision and Pattern Recognition (pp. 22500-22510).
> > >
> > >[4] Rajeswaran, Aravind, et al. "Meta-learning with implicit gradients." Advances in neural information processing systems 32 (2019).
> > >
> > >[5] Liu, Bo, et al. "Bome! bilevel optimization made easy: A simple first-order approach." Advances in Neural Information Processing Systems 35 (2022): 17248-17262.
> > >
> > >[6] Liu, Risheng, et al. "Investigating bi-level optimization for learning and vision from a unified perspective: A survey and beyond." IEEE Transactions on Pattern Analysis and Machine Intelligence 44.12 (2021): 10045-10067.

---

### Official Review · Reviewer_Sr9v · 2023-10-31

**Soundness:** 3 good
**Presentation:** 3 good
**Contribution:** 3 good
**Rating:** 6
**Confidence:** 3

**Summary:**

This paper presents a bi-level optimization approach for the unlearning of diffusion models. Specifically, the inner objective focuses on data sanitization, while the outer objective seeks to retain the utility of the diffusion model with respect to the retained data. Moreover, the proposed technique is versatile, accommodating both conditional and unconditional image generation. Its effectiveness has been demonstrated across several datasets, including UTKFace, CelebA, CelebAHQ, and CIFAR10.

**Strengths:**

1. Assess the performance of the unlearned model using a diverse set of metrics to capture multiple perspectives. These metrics include the Fréchet Inception Distance (FID), accuracy (Acc), Membership Inference Attack (MIA), Kullback-Leibler (KL) distance, and weight distance.
2. Examine the effectiveness of the proposed approach across several datasets: UTKFace, CelebA, CelebAHQ, and CIFAR10.

**Weaknesses:**

The proposed method primarily focuses on class-wise unlearning and may have limitations when applied outside this specific context (e.g., nudity removal, artistic style removal).

**Questions:**

How effective is the unlearning process when subjected to adversarial attacks?

---

> ### Author Response · Authors · 2023-11-20
> **Response to Reviewer Sr9v**
>
> Thank you for your thoughtful feedback and suggestions, which we have now incorporated into the new revision of the manuscript. **New results are included from pages 20 to 22, with changes highlighted for ease of tracking**. We address your questions below. We are happy to continue the discussion in case of follow-ups.
>
> **Q2.1 The proposed method primarily focuses on class-wise unlearning and may have limitations when applied outside this specific context (e.g., nudity removal, artistic style removal).**
>
> Our algorithm can perform unlearning outside of class-wise unlearning. Experiments on concept unlearning (unlearning the attribute 'Blond hair' or `Eyeglasses') can be found in Figure 5 (b) on page 9 in the main paper, in Figures 14 and 15 on page 18, and in Figure 16 on page 19 in the Appendix.
> Thanks, we have since added descriptions to the manuscript for better clarity.
>
> **Q2.2 How effective is the unlearning process when subjected to adversarial attacks?**
>
> Thanks for the suggestion. We conducted an additional experiment to evaluate the unscrubbed model, the retrained model, and our scrubbed model when subjected to adversarial attacks. This experiment has been added to Figure 22 on page 22. We observed that as the magnitude of the perturbation increases, the quality of the generated images decreases for all models, and the generated images by our scrubbed model still contain no information about the forgetting classes $\mathcal{C}_f$.

---

### Official Review · Reviewer_KgRz · 2023-11-08

**Soundness:** 3 good
**Presentation:** 3 good
**Contribution:** 3 good
**Rating:** 6
**Confidence:** 4

**Summary:**

**POST REBUTTAL NOTE FOR AUTHORS:**

I would like to thank the authors for patiently answering my questions and acknowledge that I have read their responses.

--------------------------------------------------

**PRE REBUTTAL REVIEW:**

This paper tackles the privacy issue with respect to the generations of diffusion models. Diffusion models, pose significant privacy risks as they can memorize and regenerate individual images from their training datasets and this paper aims to propose an unlearning algorithm. The setup considers access to a pretrained diffusion model as well as the "forgetting data"(data to be forgotten) and the "remaining data" (data which needs to be modeled correctly in the diffusion model).

EraseDiff casts this problem to a bi-level optimization problem that fine-tunes the model with the remaining data while deviating the generative process to erase the influence of the forgetting data.

**Strengths:**

- EraseDiff introduces a new approach to data unlearning in diffusion models.
- The method is more efficient than retraining the entire model, but requires more comparison in terms of other recent baselines in the literature.
- The paper provides extensive empirical evaluations, comparing with existing unlearning algorithms for neural networks.
- The technique can be applied to both conditional and unconditional image generation tasks.
- The paper grounds its methodology in a solid theoretical framework

**Weaknesses:**

- I understand the assumptions made in the paper are that we do have access to $\mathcal{D}_r$. However, is this a reasonable assumption for large scale diffusion models? Often times, we have access to a pre-trained diffusion model and also to the forgetting data ($\mathcal{D}_f$) but assuming access to $\mathcal{D}_r$ which may be very large might not be reasonable. Have the authors thought about the case where we do not have access to the remaining data? How does the algorithm change? Is there significant impact on the results?

- The impact of unlearning seems to have affected the samples quality significantly.
- The assumption in the methodology that they access to $\mathcal{D}_r$, hinders the use of the algorithm for large scale diffusion models such as Stable Diffusion.

See my questions below.

**Questions:**

- There is no definition of the hyper-parameter $\lambda$. Does it control the balance between retaining and forgetting data?

- Authors have missed important citations: [2] and [3].

- Can the authors clarify on the connections and similarity of their work compared to [1], [2] and [3]? From my understanding, [1] also fine-tunes the score network to minimize the generation probability of samples that can be labeled as a specific class. Also in [2] and [3] authors propose similar approaches for removing concepts. I believe the claim made in the introduction about this work being the first to study unlearning in diffusion models is incorrect.

- A believe it would be nice to be able to compare your method against Selective Amnesia [2]. In Table 1 of [2] there is results on CIFAR10  and the FID on the remaining classes seems to be much lower (9.08) than what you have reported in Table 2 (seems to be on average above 20). I would appreciate it if the authors provide more comparisons or clarifications with respect to [2].

- The texts on Figure 3 are not easily readable. What is the y axis (frequency)? How do we interpret this plot?

I am willing to modify my score once the results and comparisons with existing baselines in the literature are clarified.


[1] Rohit Gandikota, Joanna Materzynska, Jaden Fiotto-Kaufman, and David Bau. Erasing concepts from diffusion models. arXiv preprint arXiv:2303.07345, 2023.

[2] Heng, Alvin, and Harold Soh. "Selective Amnesia: A Continual Learning Approach to Forgetting in Deep Generative Models." arXiv preprint arXiv:2305.10120 (2023).

[3] Gandikota, R., Orgad, H., Belinkov, Y., Materzyńska, J., & Bau, D. (2023). Unified concept editing in diffusion models. arXiv preprint arXiv:2308.14761.

---

> ### Author Response · Authors · 2023-11-20
> **Response to Reviewer KgRz**
>
> Thank you for your thoughtful feedback and suggestions, which we have now incorporated into the new revision of the manuscript. **New results are included from pages 20 to 22, with changes highlighted for ease of tracking**. We address your questions below. We are happy to continue the discussion in case of follow-ups.
>
> **Q1.1 How does the algorithm change when we do not have access to the remaining data $\mathcal{D}_r$?**
>
> In response to your comment and following the methodology outlined in [2], we conducted an additional experiment using Generative Replay (GR) to generate images $\mathcal{D}_r'$ as a substitute for $\mathcal{D}_r$ in the unlearning process. The findings from this experiment are presented in our revised manuscript, specifically in Table 6 and Figure 19 on page 21 in the revised manuscript. The results indicate that while image quality diminishes post-scrubbing when using generated images, our model still significantly outperforms other methods.
>
> Additionally, we would like to highlight that in our initial submission, the unlearning process was conducted using only 8K data points, randomly selected from $\mathcal{D}_r$. Following your suggestion, we have not only clarified this in the main paper but also explored the impact of varying the number of samples from $\mathcal{D}_r$ on our algorithm, detailed in Figure 6 on page 15. We found that increasing the sample size can enhance the quality of images generated from the remaining classes, albeit at the price of slight detriment to the forgetting classes.
>
> **Q1.2 Can the authors clarify the connections and similarity of this work compared to [1-3]?**
>
> Thanks for the suggestion. All these methods aim to erase some information carried by the diffusion models.
> However, [1] and [3] mainly focus on text-to-image models and high-level visual concept erasure ([3] also focus on debiasing).
> Both the method proposed in [2] and ours aim to maximize the distance between the ground-truth backward distribution and the learnable backward distribution, to perform class-wise unlearning and concept erasure for diffusion models.
> Thanks, we have since added this in the related work section and removed the statement in the introduction.
>
> **Q1.3 Compare against Selective Amnesia [2] and FID score clarifications.**
>
> In the current stage, we follow the same setting in [2] to forget samples belonging to the label '0' on CIFAR-10 and use the well-trained model from [2]. From [2], the FID score on the $\mathcal{C}_r$ is 9.67 for the original model and 9.08 for Selective Amnesia [2].
> With an EMA decay of 0.99, our method achieves a FID score of 8.93 at 210 steps, 8.83 at 205 steps, and 8.90 at 200 steps. Generated examples are shown in Figure 20 on page 21 of the revised manuscript.
>
> Regarding the FID score on CIFAR-10, results in Table 2 are reported for DDIM with inference time step $T=100$, and FID is computed over 5K images separately for each class, while results in Table 1 of [2] are reported for DDPM with inference time step $T=1000$ and FID is computed over 5K$\times 9$ images.
> For your reference, the FID score computed over 40K images is around 8.18 in Table 2.
>
> **Q1.4 Does hyper-parameter $\lambda$ control the balance between retaining and forgetting data?**
>
> Yes, $\lambda$ controls the balance between retaining and forgetting data. An ablation study with respect to $\lambda$ is provided in Figure 6. Thanks, we have since added a description in the main text for clarity.
>
> **Q1.5 The texts on Figure 3 are not easily readable. What is the y axis (frequency)? How do we interpret this plot?**
>
> The y-axis represents the number of occurrences of the specific value.
> The diffusion process would add noise to the clean input $x$, consequently, the model output $\epsilon_T$ distribution given these noisy images $x_t$ tends to converge towards the Gaussian noise distribution $\epsilon$.
> As shown in Figure 3, we can see that the distance between the output distribution with respect to $\mathcal{C}_r$ and the Gaussian distribution for our scrubbed model is close to that of the unscrubbed model, while the distance between the output distribution with respect to the $\mathcal{C}_f$ is further away from the Gaussian distribution for our scrubbed model than the unscrubbed model, indicating that our algorithm preserves the model utility over $\mathcal{C}_r$ while successfully scrub the information concerning $\mathcal{C}_f$.
> Thanks, we have since revised the context for better clarity.

---

### Author Response · Authors · 2023-11-20
**General Response**

We appreciate the time the reviewers have dedicated to our work, and we thank all the reviewers for their constructive feedback and suggestions. Below, we summarize the main concerns raised by the reviewers, along with our efforts to address them. We will then provide answers to each individual question from the reviewers separately. In response to the comments, we have revised our work and updated the PDF file accordingly. **New results are included from pages 20 to 22, with changes highlighted for ease of tracking**.

**GQ1: Reviewer ${\color{red}KgRz}$ inquired about the performance of our algorithm in scenarios where the remaining data $\mathcal{D}_r$ is not available.**

In response to the comment, and following [2], we conducted an additional experiment.
In this scenario, the model generates the remaining data, which we denote as $\mathcal{D}_{r'}$, to unlearn $\mathcal{D}_f$. This experiment has been added to Table 6 and Figure 19 on page 21 in the revised manuscript.
We observed that, although the performance may decrease compared to scenarios where $\mathcal{D}_r$ is available, our algorithm still surpasses other methods. Our algorithm without $\mathcal{D}_r$ generates unrecognized contents for the forgetting classes $\mathcal{C}_f$ and reasonable images for the remaining classes $\mathcal{C}_r$, while other methods either cannot erase information about $\mathcal{C}_f$ or fail to generate reasonable images for $\mathcal{C}_r$.

**GQ2. Reviewers ${\color{cyan}JxPy}$ and ${\color{violet}oTvb}$ inquired about two alternative formulations to perform unlearning within our framework and their possible benefits. $\textbf{I.}$ formulate the optimization as a multi-task problem instead of a bi-level one, and $\textbf{II.}$ performing a disjoint optimization by first unlearning $\mathcal{D}_f$, followed by retaining back the knowledge by retraining on $\mathcal{D}_r$.**

Indeed, both formulations are possible, however, none would perform on par or better than our proposal empirically.
To provide further insights, we performed extra experiments on the CIFAR-10 dataset using conditional DDIM (please see Tables 4 and 5, Figures 17 and 18 on page 20 in the revised manuscript).  Formulation as multi-task problem and solving it via scalarization as a single-level optimization is shown as SO in Table 4 and Figure 17. We observed that this would lead to a significant drop in the image quality on $\mathcal{C}_r$ when boosting to forget $\mathcal{C}_f$ by large. As shown in Figure 17 ($\alpha=0.5$), when the generated images yield coarse information for $\mathcal{C}_f$, the quality of generated images for $\mathcal{C}_r$ drops significantly. Overall, we believe formulating as vanilla multi-task problem is not beneficial.

Regarding disjoint optimization, the nature of first forgetting and then retraining suggests possible degradation as either the model is seriously damaged by the forgetting step or even the possibility of relearning the forgetting data as a result of retraining (especially for the classes with large resemblances). To put this to test, we performed an experiment on CIFAR-10 by first performing NegGrad and then relearning using the remaining data. Results, denoted as Two-steps, are shown in Table 5 and Figure 18. We observe significant deterioration in generating $\mathcal{C}_r$, while improvement in scrubbing $\mathcal{C}_f$. This formulation is observed to be even worse than the multi-task formulation. Interestingly, we observe that the generated images for $\mathcal{C}_f$ may contain relevant information after retraining. Overall, our proposed formulation leads to robust and a simple optimization framework.
Further studies on effective multi-task formulation and disjoint optimization remain as future work which we have reflected in the conclusion of our revised manuscript.

---
>[1] Rohit Gandikota, Joanna Materzynska, Jaden Fiotto-Kaufman, and David Bau. Erasing concepts from diffusion models. arXiv preprint arXiv:2303.07345, 2023.
>
>[2] Heng, Alvin, and Harold Soh. "Selective Amnesia: A Continual Learning Approach to Forgetting in Deep Generative Models." arXiv preprint arXiv:2305.10120 (2023).

---

### Meta-Review · Area_Chair_pMXD · 2023-12-10

**Metareview:**

This paper was reviewed by five knowledgeable referees, who raised concerns w.r.t.:
1. The assumption of having access to Dr (all training data which is not to be forgotten) appeared unrealistic and hindered the applicability of the proposed approach to state-of-the-art diffusion models (trained on large scale datasets) (KgRz)
2. The impact of forgetting on the sample quality (KgRz, oTvb)
3. Some missing important citations and comparisons (KgRz)
4.  The applicability of the method beyond class removal (Sr9v, LbvE)
5. The effectiveness of removal when subjected to adversarial attacks (Sr9v)
6. The rationale of the proposed bi-level optimization approach which appeared unclear (JxPy, oTvb).

The rebuttal partially addressed the reviewers' concerns. In particular, the authors clarified how the proposed approach could still be applied without assuming access to Dr - by either leveraging generative replay or by subsampling Dr. The authors also extended the experimental evidence to include sample unlearning, the alternative formulations proposed by reviewers JxPy and oTvb, and addressed the concerns w.r.t. adversarial attacks. Both authors and reviewers followed up during the discussion period. Key discussion elements included the bi-level optimization formulation and the positioning of the proposed approach w.r.t. selective amnesia. After discussion, the reviewers remained hesitant about the positioning and comparisons to selective amnesia (key differences and metrics computed in different ways), and the bi-level formulation which still appeared questionable. Moreover, the scope of the results was found narrow (focused on CIFAR-10), and the reviewers expected to see results on state-of-the-art models such as stable diffusion given the relaxation on the Dr assumption. During discussion, reviewer Sr9v communicated that they would like to lower their score to 5.

The AC agrees with the final assessment/concerns of the reviewers and recommends to reject. The AC encourages the authors to consider the final recommendations of the reviewers (e.g. experimental scope, use of state-of-the-art models trained on large scale datasets, positioning and comparisons with selective amnesia) to strengthen the otherwise interesting contribution and improve the next iteration of their work.

**Justification For Why Not Higher Score:**

Although the AC considers the presented contribution interesting, some key points would need to be addressed prior to acceptance. To strengthen the contribution the authors should consider extending the experimental scope (other datasets, and models pre-trained on large scale datasets), positioning and contrasting the proposed approach with selective amnesia (and comparing following the same metric computation protocol), and improving the presentation by clarifying the motivation for a bi-level optimization.

**Justification For Why Not Lower Score:**

N/A

---

### Decision · Program_Chairs · 2024-01-16

Reject